# LRRK2 regulates endoplasmic reticulum–mitochondrial tethering through the PERK-mediated ubiquitination pathway

Toshihiko Toyofuku[1,*] (iD), Yuki Okamoto[1], Takako Ishikawa[1], Shigemi Sasawatari[1] & Atsushi Kumanogoh[2]

## Abstract

Mutations in the leucine-rich repeat kinase 2 (*LRRK2*) gene are the most common cause of familial Parkinson's disease (PD). Impaired mitochondrial function is suspected to play a major role in PD. Nonetheless, the underlying mechanism by which impaired LRRK2 activity contributes to PD pathology remains unclear. Here, we identified the role of LRRK2 in endoplasmic reticulum (ER)–mitochondrial tethering, which is essential for mitochondrial bioenergetics. LRRK2 regulated the activities of E3 ubiquitin ligases MARCH5, MULAN, and Parkin via kinase-dependent protein–protein interactions. Kinase-active LRRK2(G2019S) dissociated from these ligases, leading to their PERK-mediated phosphorylation and activation, thereby increasing ubiquitin-mediated degradation of ER–mitochondrial tethering proteins. By contrast, kinase-dead LRRK2(D1994A)-bound ligases blocked PERK-mediated phosphorylation and activation of E3 ligases, thereby increasing the levels of ER–mitochondrial tethering proteins. Thus, the role of LRRK2 in the ER–mitochondrial interaction represents an important control point for cell fate and pathogenesis in PD.

**Keywords** endoplasmic reticulum; LRRK2; mitochondria; PERK; ubiquitin ligase

**Subject Categories** Autophagy & Cell Death; Membranes & Trafficking; Metabolism

**The EMBO Journal (2020) 39: e100875**

## Introduction

Parkinson's disease (PD) is a neurodegenerative disorder with no cure. Genetic studies revealed that missense mutations in the protein LRRK2 are the most common cause of familial PD (Funayama *et al*, 2002; Paisan-Ruiz *et al*, 2004; Zimprich *et al*, 2004). In addition, genome-wide association studies have identified a common variation in the *LRRK2* gene as a risk factor for sporadic PD (Satake *et al*, 2009; Simon-Sanchez *et al*, 2009). *LRRK2* encodes a 2,527–amino acid protein consisting of an ankyrin-repeat (ANK) domain, a leucine-rich repeat (LRR), a Ras of complex proteins (ROC) domain, a C-terminal of Roc (COR) domain, and kinase and WD40 domains.

Mitochondrial dysfunction has been implicated in a range of neurodegenerative diseases and in PD in particular (Winklhofer & Haass, 2010). The molecular pathogenesis of sporadic PD and the basis of selective dopaminergic neuronal loss remain unclear. Mutations in several genes, including *SNCA* (encoding alpha-synuclein), *DJ-I, LRRK2, PINK1*, and *PRKN* (encoding Parkin), cause forms of familial PD that are clinically indistinguishable from sporadic PD (Klein & Westenberger, 2012). *PINK1* and *PRKN* encode mitochondrially located proteins that participate in mitochondrial quality control, further supporting the idea that mitochondrial dysfunction is sufficient to cause PD.

Mitochondria play major roles in multiple cellular processes, including energy metabolism, calcium homeostasis, and lipid metabolism. Mitochondria are associated with the endoplasmic reticulum (ER), with 5–20% of the mitochondrial surface apposed to ER membranes (Rizzuto *et al*, 1998; Csordas *et al*, 2006). The regions of the ER associated with mitochondria are termed mitochondria-associated ER membranes (MAMs), and these contacts facilitate a variety of signaling processes between the two organelles, including calcium (Gincel *et al*, 2001; Rizzuto *et al*, 2012) and phospholipid exchange (Rowland & Voeltz, 2012), and impact diverse physiological processes including ATP production, autophagy, protein folding, and apoptosis (Simmen *et al*, 2010; Rowland & Voeltz, 2012; Hamasaki *et al*, 2013; Kornmann, 2013). Despite the fundamental importance of these interactions to cell metabolism, the mechanisms that mediate recruitment of ER membranes to mitochondria are not fully understood. Several protein complexes have been proposed as ER–mitochondrial tethers, implying that different ER–mitochondrial tethering protein complexes may permit selective recruitment of different domains of the ER, causing the distances between physiological ER–mitochondrial contacts to vary 10–30 nm (Csordas *et al*, 2006; Rowland & Voeltz, 2012).

1 Department of Immunology and Molecular Medicine, Graduate School of Medicine, Osaka University, Suita, Japan
2 Department of Respiratory Medicine and Clinical Immunology, Graduate School of Medicine, Osaka University, Suita, Japan
   *Corresponding author. Tel: +81 0662 108381; E-mail: toyofuku@imed3.med.osaka-u.ac.jp

To maintain energy production and various cellular processes, mitochondrial protein quality control mechanisms are required to counteract the continuous accumulation of defective mitochondrial components. One such mechanism is the dynamic remodeling of mitochondrial membrane through fission and fusion (Karbowski & Youle, 2011), and the other is the ubiquitin/proteasome system, which removes damaged proteins in mitochondria and ER (Christianson & Ye, 2014; Ruggiano *et al*, 2014). The covalent attachment of ubiquitin to target proteins (substrates) is mediated by the sequential action of an E1-activating enzyme, an E2 conjugase, and an E3 ubiquitin ligase (Pickart & Eddins, 2004). E3 ligases have the ability to bind both E2 proteins (via a RING domain, Ubox, or HECT domain) and substrates. Mitochondria localized E3 ubiquitin ligases such as MARCH5, MULAN, and Parkin ubiquitinate MAM components to regulate MAM formation and mitochondrial morphology (Harder *et al*, 2004; Braschi *et al*, 2009; Lokireddy *et al*, 2012; Nagashima *et al*, 2014; Gladkova *et al*, 2018).

On the other hand, recent reports have revealed the contribution of ER stress to the pathogenesis of PD (Mercado *et al*, 2013). ER stress activates the unfolded protein response (UPR), a complex signal-transduction pathway that mediates restoration of ER homeostasis (Doyle *et al*, 2011). Under chronic ER stress, the UPR triggers cell death by apoptosis, eliminating damaged cells. In mammalian cells, the UPR is initiated by activation of three distinct types of stress sensors located at the ER membrane: two transmembrane kinases, PERK and IRE1α, and transcription factor ATF6. Immunohistochemistry of post-mortem brain tissue from PD patients revealed that the phosphorylated forms of PERK and its substrate, eukaryotic initiation factor 2 α (eIF2α), are present in dopaminergic neurons of the substantia nigra (Hoozemans *et al*, 2007). However, the mechanisms leading to ER stress in PD and the actual impact of the UPR on this disease remain unclear.

In this study, we investigated how LRRK2 is mechanistically involved in mitochondrial biogenesis. By analyzing metabolism and $Ca^{2+}$ transport in MEFs genetically engineered using the CRISPR/Cas9 system, we identified the mitochondrial ubiquitination system as a key target in LRRK2-mediated mitochondrial biogenesis and showed that LRRK2 regulates ubiquitin ligase activity via PERK under ER stress. Thus, our findings reveal a new functional link between vulnerability to ER stress and mitochondrial biogenesis in the context of PD pathophysiology.

# Results

Experiments were performed using genome-engineered mouse MEFs in which LRRK2 was deleted or replaced with either a kinase-active LRRK2 harboring the most common PD-related mutation (G2019S) or kinase-inactive LRRK2 with the D1994A mutation (Fig EV1A). Kinase assays using a synthetic substrate peptide (LRRKtide) revealed that LRRK2(D1994A) had lower activity, and LRRK2 (G2019S) had higher activity, than wild-type LRRK2 (Fig EV1B–D).

## Mitochondrial morphology

In PD neurons, un-fragmented damaged materials accumulate, possibly due to impaired vesicular trafficking to the lysosome (Abeliovich & Rhinn, 2016). Phosphoproteomics has revealed that

LRRK2 phosphorylates a subset of Rab GTPases thereby regulating intracellular endosome trafficking (Steger *et al*, 2016). Especially, the activity of Rab7 GTPase, a mediator for the late endosome–lysosome transport, is regulated by *drosophila LRRK2* homolog (Dodson *et al*, 2012). Electron micrography revealed multiple large, electron-dense materials in the cytoplasm of both $LRRK2^{-/-}$, LRRK2 (D1994A) and LRRK2(G2019S)-expressing MEFs (Fig 1A), suggesting that LRRK2 mutation impairs the lysosomal degradation of cytosolic debris through defects in trafficking of endosome to lysosome. Visualization of mitochondrial morphology using Mitotracker revealed that the proportion of fragmented mitochondria was elevated in $LRRK2^{-/-}$ and LRRK2(G2019S)-expressing MEFs (Fig EV1E). Consistent with the differences in mitochondrial morphology among MEFs, the activity of citrate synthase, the initial enzyme of the tricarboxylic acid (TCA) cycle and an exclusive marker of the mitochondrial matrix, was reduced in $LRRK2^{-/-}$ and LRRK2(G2019S)-expressing MEFs but elevated in LRRK2(D1994A)-expressing MEFs (Fig 1B). Thus, LRRK2 mutations disturbed mitochondrial biogenesis and/or proteasomal degradation processing.

## Mitochondrial oxidative phosphorylation

To determine the role of LRRK2 in mitochondrial energetics, we measured basal and maximal (i.e., uncoupled with FCCP) oxygen consumption rate (OCR), an indicator of mitochondrial OXPHOS, and the extracellular acidification rate (ECAR), an indicator of aerobic glycolysis in living MEFs (Figs 1C and D, and EV1F and G). Maximal OCRs were significantly lower in LRRK2(G2019S)-expressing and $LRRK2^{-/-}$ MEFs, but higher in LRKK2(D1994A)-expressing MEFs, indicating that OXPHOS was enhanced by LRRK2(D1994A) but suppressed by loss of LRRK2 or expression of LRRK2(G2019S) (Fig 1C and D). By contrast, ECAR did not differ significantly between MEFs expressing LRRK2 mutants (Fig EV1F), indicating that aerobic glycolysis was not altered by LRRK2 mutation. Relative utilization of OXPHOS and glycolysis, as indicated by the OCR/ECAR ratio, was higher in LRRK2(D1994A)-expressing MEFs and lower in $LRRK^{-/-}$ and LRRK2(D2019S)-expressing MEFs (Fig EV1G). These results suggested that OXPHOS is regulated by LRRK2 in a kinase-dependent manner.

The reduced rate of OXPHOS in LRRK2(G2019S)-expressing MEFs could be due to less active mitochondria, a lower density of mitochondria, or a combination of both. Because the rate of OXPHOS predicts ATP production, we next estimated the relative contribution of mitochondrial OXPHOS to ATP production (Fig 1E). To this end, we measured intracellular basal ATP content in MEFs in the absence and presence of oligomycin, a specific inhibitor of the mitochondrial F1F0-ATP synthase, to confirm the involvement of OXPHOS as the source of ATP production. MEFs expressing $LRRK2^{-/-}$ or LRRK2(G2019S) had significantly lower oligomycin-sensitive ATP content than wild-type MEFs. Thus, the reduction in OXPHOS activity due to LRRK2(G2019S) resulted in a decrease in ATP production.

## Autophagy

To determine whether LRRK2 regulates autophagy, we measured the LC3-II level in MEFs (Fig 1F). Under basal conditions, as well as under ER stress induced by tunicamycin, the LC3-II level was higher

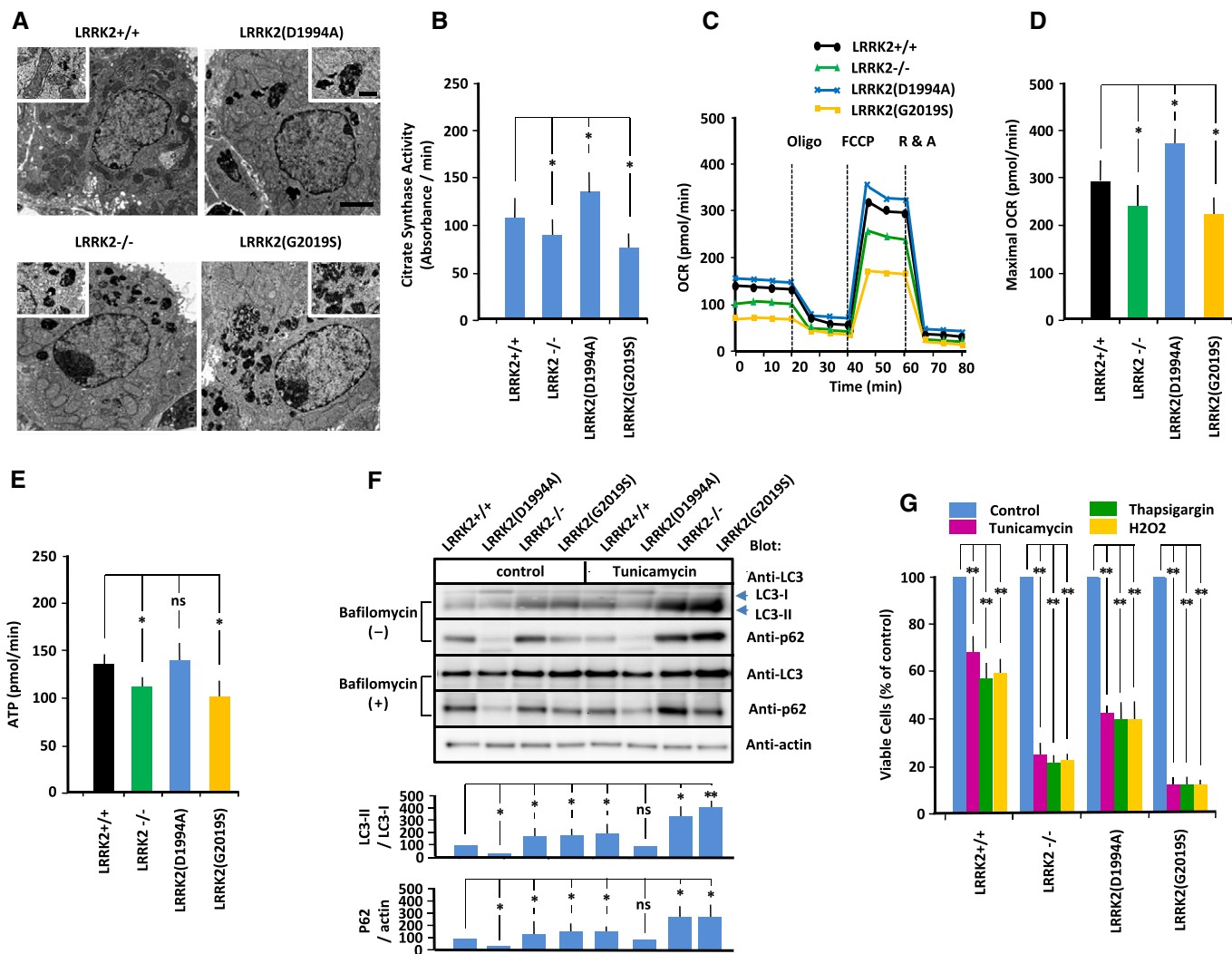

**Figure 1. LRRK2 regulates mitochondrial energetics and cellular vulnerability to ER stress.**

A   Electron micrographs of MEFs of the indicated genotypes. Scale bar: 1 μm. High-magnification images depicting representative electron-dense materials are shown. Scale bar: 0.2 μm.

B   Citrate synthase activity (absorbance/min) was measured in MEFs of the indicated genotypes. Error bars represent ± SD from eight independent experiments.

C, D   Oxygen consumption rates (OCRs) of MEFs of the indicated genotypes were measured on an XF24 Analyzer. (C) Oxygen consumption profiles for MEFs of the indicated genotypes exposed sequentially to oligomycin (2 μg/ml) (Oligo), FCCP (2 μM), and rotenone (1 μM) plus actinomycin (2 μM) (R & A). (D) Maximal OCR (pmol $O_2$/min) (*n* = 8). Error bars represent ± SD from eight independent experiments.

E   ATP production (pmol/min) (*n* = 8). Error bars represent ± SD from eight independent experiments.

F   Representative immunoblot of LC3 and p62 in MEFs of indicated genotype. MEFs were treated with tunicamycin (5 μg/ml) or vehicle control in the presence or absence of bafilomycin A1 (400 nM), and endogenous LC3 and p62 levels were measured by immunoblotting. Data represent the ratios of LC3-II to LC3-I and p62 to actin in the absence of bafilomycin, which were normalized against the corresponding values in LRRK$^{+/+}$ MEFs. Error bars represent ± SD from four independent experiments.

G   Survival rate of MEFs treated with tunicamycin (5 μg/ml), thapsigargin (1 μM), or hydrogen peroxide (100 μM). Error bars represent ± SD from four independent experiments.

Data information: For graphs (B and D–G), the *P* values were determined by a Mann–Whitney *U*-test. ns = not significant, *$P < 0.05$, **$P < 0.01$.
Source data are available online for this figure.

---

in LRRK2$^{-/-}$ and LRRK2(S2019S)-expressing MEFs than in LRRK2$^{+/+}$ MEFs, but lower in LRRK2(D1994A)-expressing MEFs. When LC3-II degradation was blocked with 200 nM bafilomycin A1 (BFA), a specific inhibitor of autophagic degradation, the higher LC3-II levels in LRRK2$^{-/-}$ and LRRK2(S2019S)-expressing MEFs and the lower LC3-II levels in LRRK2(D1994A)-expressing MEFs than in that in

LRRK2$^{+/+}$ MEFs were also detected. Thus, the lower LC3-II levels in LRRK2(D1994A)-expressing MEFs indicated that autophagosome formation was suppressed. By contrast, the higher LC3-II levels in LRRK2$^{-/-}$ and LRRK2(G2019S)-expressing MEFs indicated that autophagosome formation was enhanced. Consistent with the results for LC3-II, the levels of p62, another substrate of autophagy, were

reduced in LRRK2(D1994A)-expressing MEFs, but elevated in LRRK2$^{-/-}$ and LRRK2(S2019S)-expressing MEFs. Thus, LRRK2-mutant MEFs exhibited impaired autophagic flux, as previously demonstrated (Alegre-Abarrategui *et al*, 2009; MacLeod *et al*, 2013).

### Cell survival under ER stress

To determine whether LRRK2 regulates cellular vulnerability to ER stress, we performed MTT assays to measure the viability response to ER stress inducers such as tunicamycin and thapsigargin, as well as oxidative stressors such as hydrogen peroxide (Fig 1G). Treatment with inducers of ER stress or oxidative stress decreased cell viability more strongly in LRRK2$^{-/-}$ and LRRK2(G2019S)-expressing MEFs than in MEFs expressing other LRRK2 mutants. Thus, LRRK2 (G2019S) exhibited greater vulnerability to ER and oxidative stresses. Together, these biochemical analyses indicated that LRRK2 regulates the viability response to ER stress, whereas kinase-active LRRK2(G2019S) enhances cellular vulnerability to this type of stress.

### Calcium homeostasis

Key enzymes of OXPHOS, such as F1F0-ATPase and pyruvate dehydrogenase, are regulated by mitochondrial matrix Ca$^{2+}$ (Territo *et al*, 2000; Balaban *et al*, 2005). Autophagy has been implicated in the IP3R-mediated mechanism (Sarkar *et al*, 2005; Criollo *et al*, 2007; Vicencio *et al*, 2009) and is activated by defects in IP3-induced Ca$^{2+}$ release (Cardenas *et al*, 2010). Cellular vulnerability to stress is associated with mitochondrial mishandling of Ca$^{2+}$ (Orrenius *et al*, 2003). These findings suggested that changes in mitochondrial biogenesis of MEFs expressing mutant LRRK2 could be due to a defect in a mitochondrial Ca$^{2+}$-dependent mechanism. Accordingly, we examined the Ca$^{2+}$ machinery on both the ER and mitochondrial sides. Specifically, we measured Ca$^{2+}$ transfer from ER to mitochondria by monitoring bradykinin-stimulated calcium release from ER.

### Mitochondrial calcium transfer

To monitor mitochondrial Ca$^{2+}$ concentration ([Ca$^{2+}$]$_m$), we targeted the protein-based Ca$^{2+}$ indicator cameleon to mitochondria and then continuously visualized free Ca$^{2+}$ in the mitochondrial matrix using fluorescence energy transfer (FRET; Miyawaki *et al*, 1997, 1999). Data in the figures are presented as absolute Ca$^{2+}$ concentrations (Fig 2A–D). On average, basal [Ca$^{2+}$]$_m$ in all MEFs was similar levels (2–3 μM). Treatment with 2.5 μM bradykinin significantly increased the [Ca$^{2+}$]$_m$ transient in wild-type MEFs; the level was higher than that in LRRK2$^{-/-}$ and LRRK2(G2019S)-expressing MEFs, but lower than that in LRKK2(D1994A)-expressing MEFs (Fig 2B). Thus, mitochondrial Ca$^{2+}$ transfer is inactivated by LRRK2$^{-/-}$ and LRRK2(G2019S), but activated by LRRK2(D1994A). By contrast, treatment with bradykinin significantly decreased ER Ca$^{2+}$ concentration ([Ca$^{2+}$]$_{ER}$), as measured by the protein-based Ca$^{2+}$ indicator ER-D1 targeted to the ER, in wild-type MEFs; the level was lower than that in LRRK2$^{-/-}$ and LRRK2(G2019S)-expressing MEFs (Fig EV2A and B). Thus, the magnitude of the change in [Ca$^{2+}$]$_m$ was reciprocal with that of the change in [Ca$^{2+}$]$_{ER}$.

Close proximity between ER-localized IP3R and OMM-localized VDAC1 at the MAM potentiates rapid transfer of Ca$^{2+}$ through the OMM. Mitochondrial Ca$^{2+}$ accumulation is augmented by IP3-activated IP3R (Rizzuto *et al*, 1993) or over-expression of VDAC1 (Madesh & Hajnoczky, 2001; Rapizzi *et al*, 2002), but attenuated by down-regulation of either protein. To determine whether IP3R or VDAC1 is involved in disrupting mitochondrial Ca$^{2+}$ accumulation in LRRK-modified MEFs, we measured [Ca$^{2+}$]$_m$ in MEFs in which IP3R or VDAC1 was modified. shRNA-mediated down-regulation of IP3R or pretreatment with 20 μM 2-APB, a membrane-permeable blocker of IP3R, attenuated [Ca$^{2+}$]$_m$ in all MEFs (Fig 2C), confirming the crucial role of IP3R in mitochondrial Ca$^{2+}$ transfer. Over-expression of IP3R increased peak [Ca$^{2+}$]$_m$ in LRRK2 (D1994A)-expressing MEFs but not in LRRK2$^{-/-}$ or LRRK2 (G2019S)-expressing MEFs. Down-regulation of VDAC1 decreased peak [Ca$^{2+}$]$_m$ in all MEFs, whereas over-expression of VDAC1 increased peak [Ca$^{2+}$]$_m$ in LRRK2(D1994A)-expressing MEFs but not in LRRK2$^{-/-}$ or LRRK2(G2019S)-expressing MEFs (Fig 2D). Thus, Ca$^{2+}$ transfer through IP3R and VDAC1 was suppressed by LRRK2(G2019S), but enhanced by LRRK2(D1994A).

### Physical interaction and Ca$^{2+}$ transfer between ER and mitochondria

To obtain insight into the mechanism by which LRRK2 influences ER–mitochondrial Ca$^{2+}$ transfer, we analyzed the relationship between the ER and mitochondria. Specifically, we performed ultrastructural analysis by electron microscopy to evaluate ER–mitochondrial contacts (Figs 2E and EV2C). Visual inspection of EM images acquired by facility personnel blinded to sample identity revealed a reduction in the number of ER–mitochondrial contact sites per unit of mitochondrial perimeter in LRRK2$^{-/-}$ and LRRK2(G2019S)-expressing MEFs. Thus, LRRK2 ablation and LRRK2(G2019S) block ER–mitochondrial contacts.

Next, we examined the physical interaction between the ER and mitochondria by *in situ* proximity ligation assay (PLA) using two organelle-surface proteins involved in the calcium channeling complex: IP3R and VDAC1 at the MAM interface (Fig 2F; De Vos *et al*, 2012; Hedskog *et al*, 2013). IP3R and VDAC1 were in close proximity in wild-type MEFs, in which PLA intensity was higher than that in LRRK2$^{-/-}$ and LRRK2(G2019S)-expressing MEFs, but lower than that in LRKK2(D1994A)-expressing MEFs. Thus, LRRK2 might be involved in MAM formation in a kinase-dependent manner.

To determine whether reduced ER–mitochondrial Ca$^{2+}$ transfer in LRRK2$^{-/-}$ and LRRK2(G2019S)-expressing MEFs was indeed due to a decrease in ER–mitochondrial contacts, we performed a rescue experiment using a synthetic ER–mitochondrial tethering protein, TOM-mRFP-ER)(Csordas *et al*, 2006; Kornmann, 2013), which restores changes in PLA intensity in LRRK2$^{-/-}$ and LRRK2(G2019S)-expressing MEFs (Figs 2G and EV2E). Over-expression of TOM-mRFP-ER significantly rescued mitochondrial Ca$^{2+}$ transfer in LRRK2$^{-/-}$ and LRRK2(G2019S)-expressing MEFs (Fig 2G). These results indicated that ER–mitochondrial tethering was suppressed by loss of LRRK2 or expression of LRRK2(G2019S).

### ER–mitochondrial tethering proteins

MAM integrity depends on the interaction of ER–mitochondrial tether proteins, whose levels are regulated by the ubiquitin/proteasome pathway (Karbowski & Youle, 2011; Christianson & Ye, 2014;

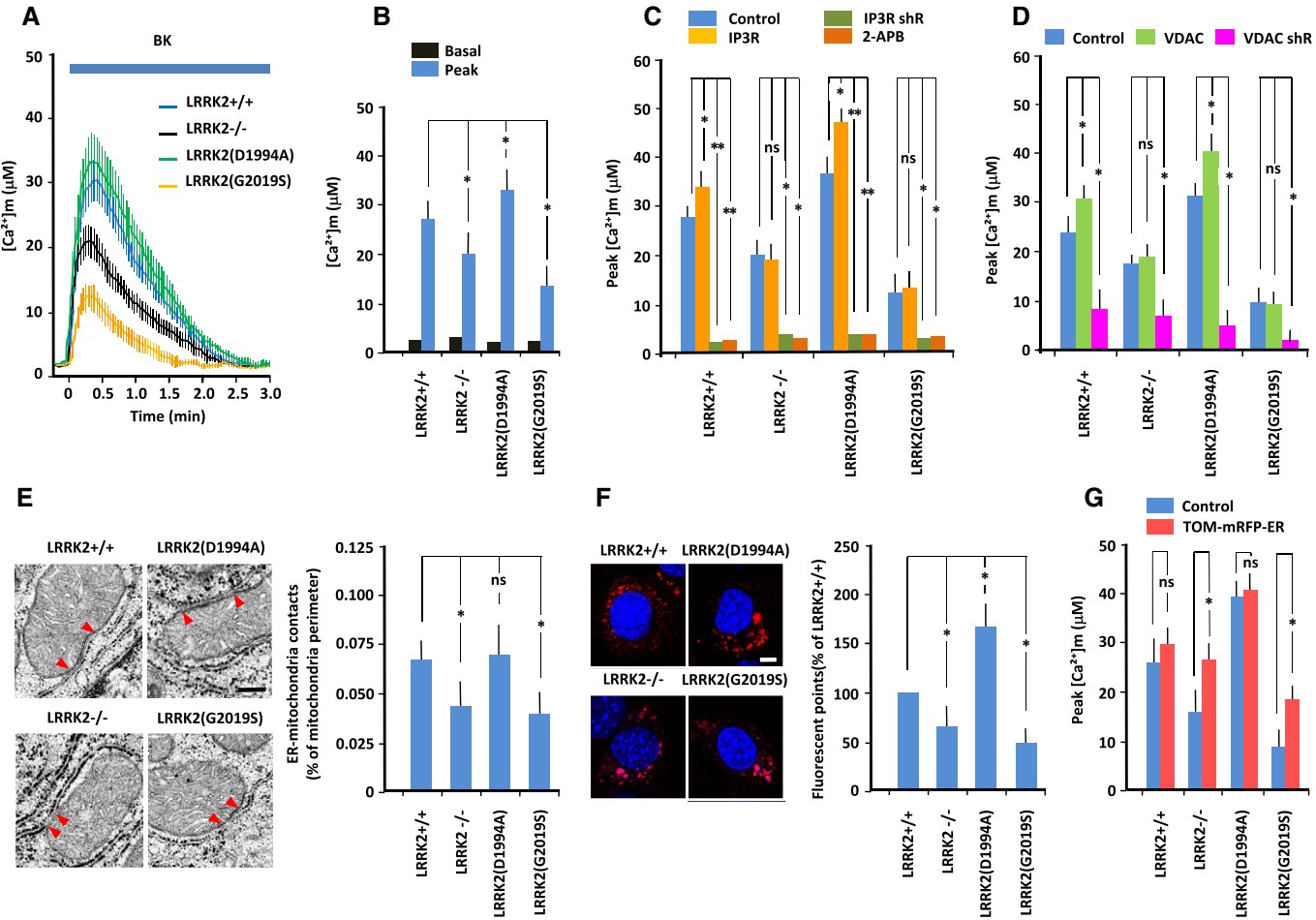

**Figure 2. LRRK2 regulates ER–mitochondrial Ca²⁺ transfer and tethering.**

A–D  MEFs were transfected with mitochondrially targeted cameleon. Free $Ca^{2+}$ dynamics in the mitochondrial matrix were visualized using FRET. Mitochondrial $[Ca^{2+}]$ ($[Ca^{2+}]_m$) was continuously monitored by FRET imaging; data are represented as absolute $[Ca^{2+}]$ in μmol. (A) Absolute $[Ca^{2+}]_m$ changes in MEFs of indicated genotype in response to bradykinin (2.5 μM). (B) Basal and peak values of $Ca^{2+}$ transients (μM). (C) Peak values of $Ca^{2+}$ transients in MEFs transfected with IP3R or shRNA against IP3R, or treated with 2-AP (20 mM). (D) Peak values of $Ca^{2+}$ transients in MEFs transfected with VDAC1 or shRNA against VDAC1. Error bars represent ± SD from six independent experiments.

E  Representative electron micrographs of MEFs of the indicated genotypes. Full arrowheads designate the limits of the zone of intimate contact between ER and mitochondria (< 20 nm). Scale bar: 100 nm. Data represent percentage of ER–mitochondrial contacts per unit of mitochondrial perimeter. Error bars represent ± SD from 40 images from MEFs of the indicated genotypes.

F  *In situ* PLA images using anti-IP3R and anti-VDAC1 antibodies. Scale bar: 20 μm. Data represent the number of fluorescent puncta in MEFs of the indicated genotypes, normalized against the value for LRRK2^(+/+) MEFs. Error bars represent ± SD from six independent experiments.

G  Peak values of $Ca^{2+}$ transients in MEFs transfected with synthetic tethering protein (TOM-mRFP-ER) to induce artificial tethering of the ER and mitochondria. Error bars represent ± SD from six independent experiments.

Data information: For graphs (B–G), the *P* values were determined by a Mann–Whitney *U*-test. ns = not significant, *$P < 0.05$, **$P < 0.01$.

Marchi *et al*, 2014; Ruggiano *et al*, 2014). It is plausible that the catalytic activities of E3 ubiquitin ligases in MEFs expressing kinase-active LRRK2(G2019S) could change the levels of MAM components in such a manner as to diminish ER–mitochondrial $Ca^{2+}$ transfer. To explore this possibility, we analyzed the expression levels of each component of isolated MAM fractions by immunoblot (Fig 3A, Appendix Fig S1A). Among the components we analyzed, those involved in the $Ca^{2+}$ transfer pathway including IP3R, VDAC1 and GRP75, and ER membrane proteins including Bap31, VAPB and Formin 2 were unchanged in all MEFs. Levels of ER and mitochondrial membrane proteins such as mitofusins 1 and 2, and

mitochondrial membrane proteins including Fis1 and PTPIP51 were lower, and DRP1 was higher, in LRRK2^(−/−) and LRRK2(G2019S)-expressing MEFs than in wild-type MEFs, whereas mitofusins 1 and 2 were present at higher levels in LRKK2(D1994A)-expressing MEFs. MAM provides a platform for mitochondrial dynamics, including DRP1-mediated fission and mitofusin 1/2-mediated fusion (Youle & van der Bliek, 2012). Thus, changes in the relative abundances of MAM components, specifically, reductions in the levels of mitofusins 1 and 2 and an increase in the level of DRP1, caused greater mitochondrial fragmentation in MEFs expressing LRRK2(G2019S) (Fig EV1E). Impaired ER–mitochondrial contact caused by changes

in MAM components could lead to changes in mitochondrial $Ca^{2+}$ transfer.

### E3 ubiquitin ligases involved in the ER–mitochondrial interaction

To determine which domain of LRRK2 was responsible for the ER–mitochondrial $Ca^{2+}$ transfer, we introduced deletion constructs of LRRK2(G2019S) into LRRK2$^{-/-}$ MEFs (Fig 3B and C, Appendix Fig

S1B). A construct lacking the N-terminal region containing the ANK, LRR, and COR domains rescued the decreased $Ca^{2+}$ transfer observed in LRRK2$^{-/-}$ MEFs, suggesting that the N-terminal domain (a.a. 1–1,515) contains the regulatory site for LRRK2. Next, we searched the binding proteins with the yeast two-hybrid system, using the N-terminal region of LRRK2 as bait and a mouse brain cDNA library as prey. Among the potential binding proteins, we identified E3 ubiquitin ligases, including MARCH5, MULAN, and

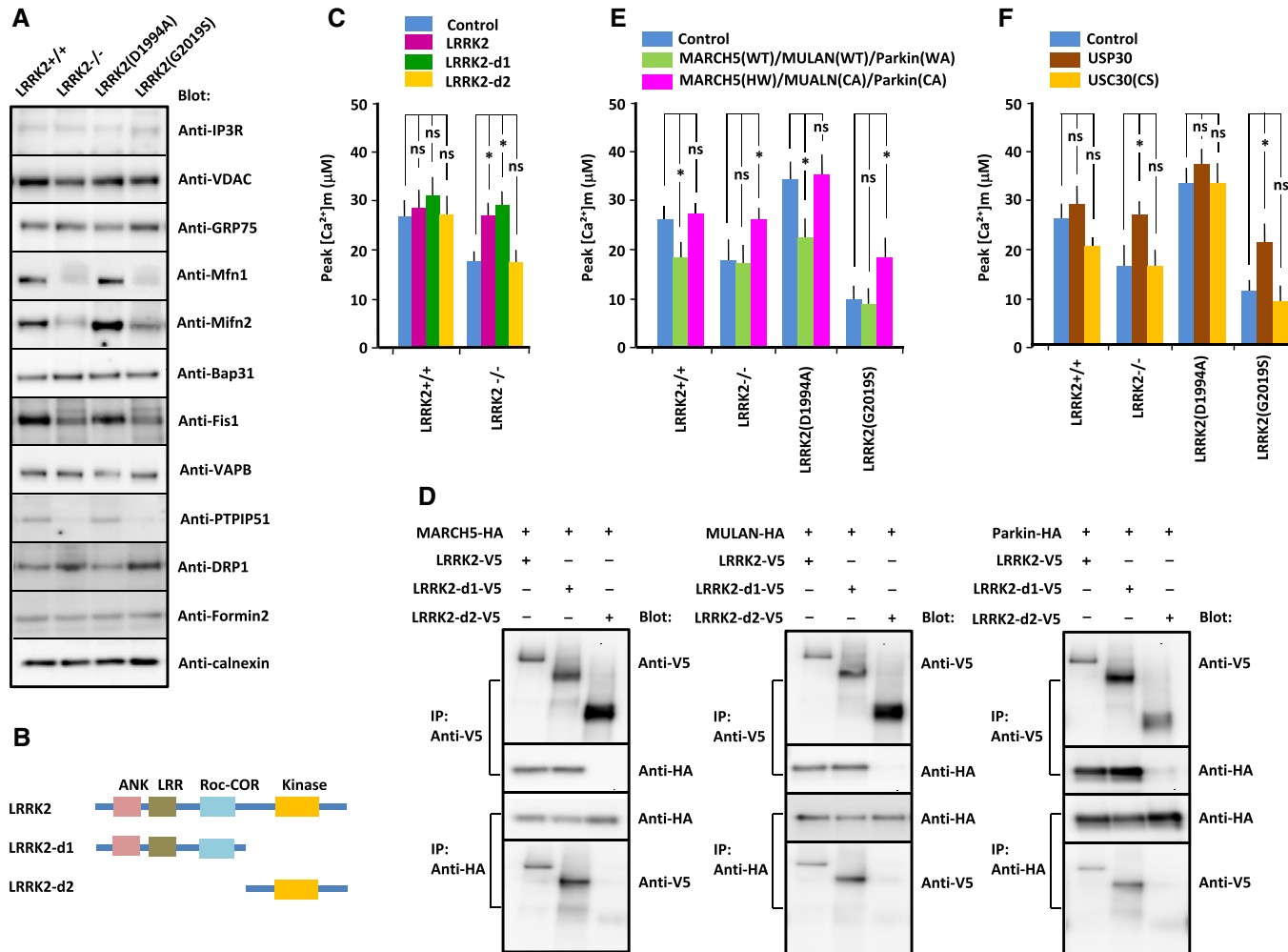

**Figure 3. Physical and functional interactions between LRRK2 and E3 ubiquitin ligases.**

A   Representative immunoblots of MAM components in MEFs of indicated genotype. MAM fraction was extracted from MEFs by the Percoll gradient method and immunoblotted with antibodies indicated at the right.
B   Diagram showing full-length and deletion constructs of LRRK2(G2019S). ANK, Ankyrin-repeat domain; LRR, leucine-rich domain; ROC, Ras complex domain; COR, C-terminus of Roc domain; Kinase, kinase domain.
C   Peak values of $Ca^{2+}$ transients in MEFs of the indicated genotypes transfected with deletion constructs of LRRK2(G2019S). Error bars represent ± SD from six independent experiments.
D   Binding assays to detect interactions between deletion constructs of LRRK2(G2019S) and E3 ubiquitin ligases. HEK293 cells were transfected with deletion constructs of LRRK2(G2019S) and E3 ubiquitin ligases, and cell lysates were immunoprecipitated with antibody against the V5 or HA epitope. Precipitated proteins were subjected to SDS/PAGE, and blots were stained with antibody against the V5 or HA epitope, as indicated to the right of each panel.
E, F   Peak values of $Ca^{2+}$ transients in MEFs of the indicated genotypes transfected with ligase-active MARCH5(WT), MULAN(WT), and Parkin(W403A) [Parkin(WA)] or dominant-negative forms of MARCH5(H43W) [MARCH(HW)], MULAN(C339A) [MULAN(CA)], and Parkin(C431A) [Parkin(CA)] (E) or in MEFs transfected with active USP30 or inactive USP30(C77S) [USP(CS)] (F). Error bars represent ± SD from six independent experiments.

Data information: For graphs (C, E and F), the *P* values were determined by a Mann–Whitney *U*-test. ns = not significant, *$P < 0.05$.
Source data are available online for this figure.

Parkin. All of these molecules belong to the really interesting new genes (RING) domain E3 ubiquitin ligase family, which is characterized by the presence of the RING domain (Deshaies & Joazeiro, 2009). During the ubiquitination process, E3 ubiquitin ligase binds to the E2-co-enzyme via its RING domain and it physically receives the ubiquitin moiety on its active center (Caulfield et al, 2015). Immunoprecipitation/immunoblot analysis confirmed the binding of each of these molecules to the N-terminal domain of LRRK2 (Fig 3D).

Several lines of evidence indicate the importance of ubiquitination and proteasomal degradation in MAM formation (Karbowski & Youle, 2011; Christianson & Ye, 2014; Marchi et al, 2014; Ruggiano et al, 2014). Mitofusin 2, a critical component in MAM formation, is ubiquitinated and degraded by ligase-active E3 ubiquitin ligases such as MARCH5(WT), MULAN(WT), or Parkin(W403A), but not by ligase-negative E3 ubiquitin ligases such as MARCH5(H43W), MULAN(C339A), or Parkin(C431A) (Gegg et al, 2010; Sugiura et al, 2013; Yun et al, 2014). Therefore, we considered it likely that LRRK2 regulates ER–mitochondrial interaction through bound E3 ubiquitin ligases. To determine whether bound E3 ubiquitin ligases are involved in LRRK2-mediated ER–mitochondrial $Ca^{2+}$ transfer, we introduced the ligase-active or ligase-dead forms of each molecule into $LRRK2^{-/-}$ and LRRK2(G2019S)-expressing MEFs, and analyzed ER–mitochondrial $Ca^{2+}$ transfer in the transfected cells (Fig 3E, Appendix Fig S1C). The combination of ligase-active MARCH5(WT), MULAN(WT), and Parkin(W403A) decreased $Ca^{2+}$ transfer in LRRK2(D1994A)-expressing MEFs, but not in $LRRK2^{-/-}$ or LRRK2(G2019S)-expressing MEFs, whereas the combination of ligase-dead MARCH5(H43W), MULAN(C339A), and Parkin(C431A) increased $Ca^{2+}$ transfer in $LRRK2^{-/-}$ and LRRK2(G2019S)-expressing MEFs, but not in LRRK2(D1994A)-expressing MEFs. Thus, ligase-negative E3 ubiquitin ligases suppressed the endogenous E3 ubiquitin ligases in the dominant-negative manner, by which over-expressed mutant E3 ubiquitin ligases may compete with endogenous E3 ubiquitin ligases for endogenous E2-co-enzyme (Caulfield et al, 2015). These results indicated that ligases were more active in $LRRK2^{-/-}$ and LRRK2(G2019S)-expressing MEFs than in LRRK2 (D1994A)-expressing MEFs. Collectively, these results suggested that the ER–mitochondrial interaction is regulated by LRRK2 through a MARCH5-, MULAN-, and Parkin-mediated mechanism involving the ubiquitin/proteasome system, in which the activities of these E3 ubiquitin ligases are promoted by loss of LRRK2 or kinase-active LRRK2(G2019S), but suppressed by kinase-dead LRRK2(D1994A).

**The role of ubiquitin deubiquitylase**

Ubiquitination of mitochondrial proteins is a reversible process in which ubiquitin is not only conjugated to substrates via the ubiquitin pathway, but also removed from substrates by deubiquitinating enzyme (Livnat-Levanon & Glickman, 2011). USP30, a mitochondrially tethered deubiquitylase, antagonizes MULAN as well as Parkin (Bingol et al, 2014; Cunningham et al, 2015). If E3 ubiquitin ligase-mediated ubiquitination is enhanced in LRRK2 (G2019S)-expressing MEFs, active USP30, but not catalytically inactive USP30 (USP30(C77S)), should antagonize the high-ubiquitination state. To confirm the observation that the perturbation of ER–mitochondrial $Ca^{2+}$ transfer in LRRK2(G2019S)-expressing

MEFs resulted from enhanced E3 ubiquitin ligase activity, we over-expressed wild-type USP30 or the C77S mutant (Fig 3F). Wild-type USP, but not USP30(C77S), partially rescued ER–mitochondrial $Ca^{2+}$ transfer in $LRRK2^{-/-}$ and LRRK2(G2019S)-expressing MEFs, but not in LRRK2(D1994A)-expressing MEFs. These results indicated that ubiquitination activity was higher in $LRRK2^{-/-}$ and LRRK2(G2019S)-expressing MEFs than in LRRK2 (D1994A)-expressing MEFs. This finding supported the idea that the activities of E3 ubiquitin ligases such as MARCH5, MULAN, and Parkin are promoted by loss of LRRK2 and kinase-active LRRK2(G2019S).

**The role of LRRK2 kinase activity in E3 ubiquitin ligase**

Does LRRK2 regulate E3 ubiquitin ligase-mediated ubiquitination and degradation of MAM components via its kinase activity? Multiple reports have shown that the activities of E3 ubiquitin ligases are regulated by phosphorylation (Gallagher et al, 2006; Smith et al, 2009; Lewandowski & Piwnica-Worms, 2014). Immunoblots of MEFs using anti-phosphoserine antibody detected higher levels of the phosphorylated forms of endogenous E3 ubiquitin ligases in LRRK2(G2019S)-expressing MEFs in the presence of ER stressors such as tunicamycin (Fig 4A). This implied that phosphorylation of E3 ubiquitin ligases is involved in their activation. To explore this possibility, we focused on ubiquitination of mitofusin 2, which was present at reduced levels in $LRRK2^{-/-}$ and LRRK2(G2019S)-expressing MEFs (Fig 3A, Appendix Fig S1A). We performed immunoblots of LRRK2-expressing MEFs transfected with each of the E3 ubiquitin ligases, cultured in the presence or absence of tunicamycin (Fig 4B). In wild-type MEFs, tunicamycin significantly increased the levels of phosphorylated E3 ubiquitin ligases, decreased the level of mitofusin 2, and reciprocally increased the level of ubiquitinated mitofusin 2. These results indicated that phosphorylation of E3 ubiquitin ligase increases its activity toward mitofusin 2. By contrast, under tunicamycin treatment, LRRK2(D1994A)-expressing MEFs contained less phosphorylated E3 ubiquitin ligase and ubiquitin-conjugated mitofusin 2, and reciprocally more mitofusin 2, than wild-type MEFs. These results indicated that LRRK2(D1994A) suppressed phosphorylation of E3 ubiquitin ligase, and the subsequent ubiquitination and degradation of mitofusin 2, under ER stress. On the other hands, $LRRK2^{-/-}$ and LRRK2(G2019S)-expressing MEFs had more phosphorylated E3 ubiquitin ligase and ubiquitin-conjugated mitofusin 2, and reciprocally less mitofusin 2, under both control conditions and tunicamycin treatment. These results indicated that loss of LRRK2 or LRRK2(G2019S) promoted phosphorylation of E3 ubiquitin ligase and subsequent ubiquitination and degradation of mitofusin 2.

To confirm this finding, we pretreated LRRK2 mutant-expressing MEFs with the LRRK2 kinase inhibitor LRRK2-IN-1 (Deng et al, 2011; Figs 4B and EV3B). In LRRK2(G2019S)-expressing MEFs pretreated with LRRK2-IN-1, E3 ubiquitin ligase phosphorylation, and the level of mitofusin 2 in the MAM fraction recovered to the levels in wild-type MEFs. In line with this finding, ER–mitochondrial $Ca^{2+}$ transfer in LRRK2-IN-1-treated LRRK2(G2019S)-expressing MEFs increased to the level in wild-type MEFs (Fig 5A). Collectively, these results indicated that LRRK2 regulates E3 ubiquitin ligase-mediated ubiquitination and proteasomal degradation in a kinase-dependent manner.

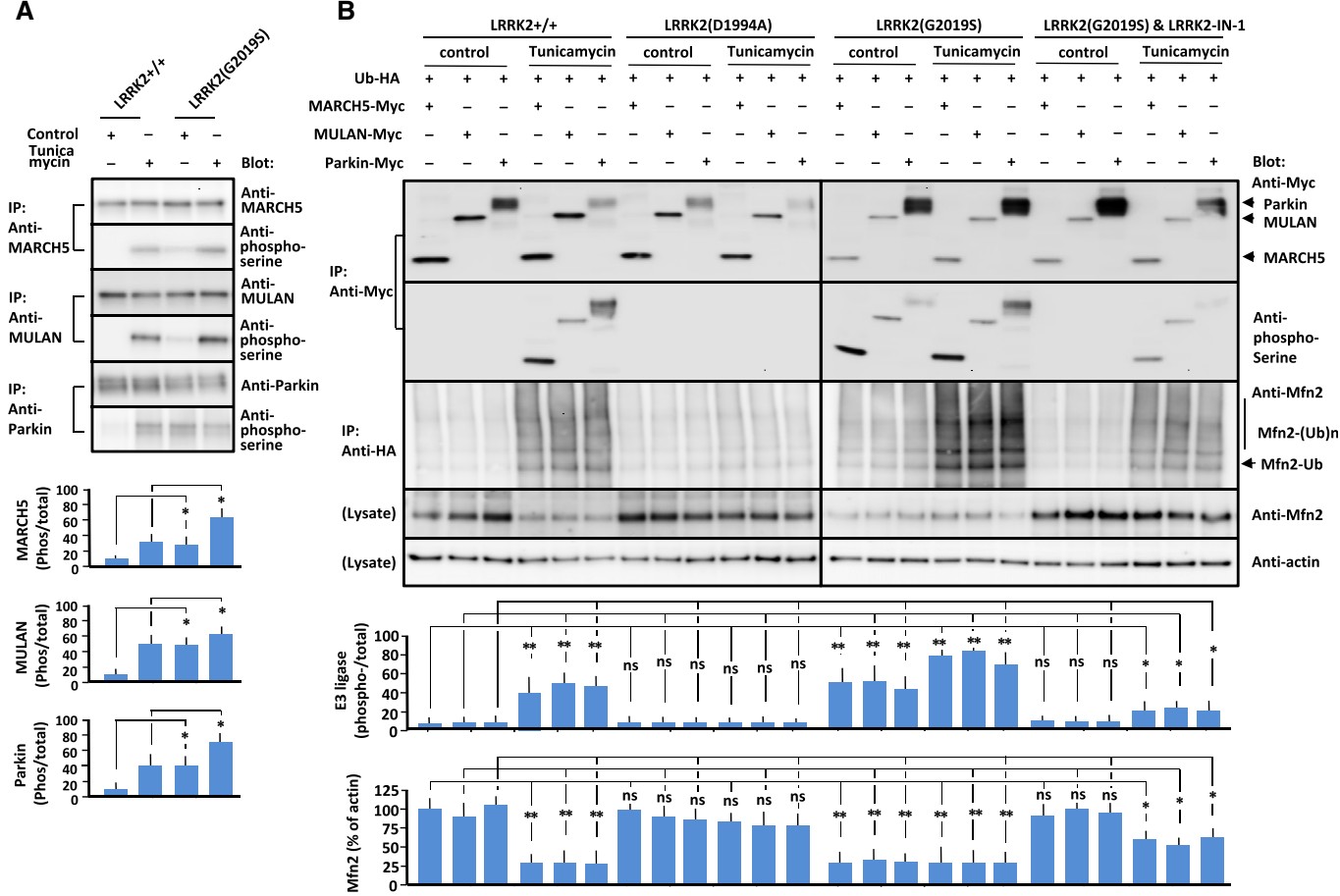

**Figure 4. Phosphorylation and activation of E3 ubiquitin ligases by LRRK2.**

A  Immunoprecipitation/Immunoblot of phosphorylated E3 ubiquitin ligases in LRRK2+/+ and LRRK2(G2019S)-expressing MEFs treated with vehicle (Control) or tunicamycin (1 μg/ml). Endogenous PERK, MARCH5, MULAN, and Parkin were immunoprecipitated with the corresponding antibody, and precipitates were immunoblotted with antibody indicated at the right. Data represent the ratio of phosphorylated to total protein levels. Error bars represent ± SD from four independent experiments.

B  Immunoprecipitation/immunoblot of phosphorylated E3 ubiquitin ligases and mitofusin 2 in MEFs of the indicated genotypes in the presence or absence of tunicamycin (5 μg/ml). LRRK2(G2019S) MEFs were also pretreated with LRRK2-IN-1 (1 μM). MEFs of the indicated genotypes were transfected with each E3 ubiquitin ligase and ubiquitin, and lysates were immunoprecipitated with antibody against the Myc or HA epitope. Precipitated proteins were subjected to SDS/PAGE, and blots were stained with antibody against the Myc epitope, phosphoserine, or mitofusin 2 as indicated to the right of each panel. Mfn2: mitofusin 2, Ub: ubiquitin. Data represent ratios of phosphorylated to total E3 ubiquitin ligase and mitofusin 2 to actin. Error bars represent ± SD from four independent experiments.

Data information: For graphs (A and B), the P values were determined by a Mann–Whitney U-test. ns = not significant, *P < 0.05, **P < 0.01.
Source data are available online for this figure.

## PERK phosphorylates E3 ubiquitin ligase

Kinase activity of LRRK2 was required for phosphorylation and activation of E3 ubiquitin ligase. However, *in vitro* phosphorylation assays did not show direct phosphorylation of E3 ubiquitin ligase by LRRK2 (data not shown), indicating that kinases other than LRRK2 phosphorylate these ligases.

To identify the kinase responsible for phosphorylation of E3 ubiquitin kinase, we introduced an siRNA library targeting the expression of 628 kinases into LRRK2(G2019S)-expressing MEFs under tunicamycin treatment. If the responsible kinase is knocked down, un-phosphorylated and inactive E3 ubiquitin ligases might rescue the cell viability of LRRK2(G2019S)-expressing MEFs treated with tunicamycin. The cell viability of MEFs in each well was

measured using the Vybrant MTT Cell Proliferation Assay. Using a statistical Z-score to quantify the deviation of cell viability from the mean of all measurements, we selected four siRNAs that induced significantly higher viability of LRRK2(G2019S)-expressing MEFs under tunicamycin treatment. Next, we performed immunoblots with anti-phospho-serine antibody to examine the phosphorylation of E3 ubiquitin ligases in LRRK(G2019S)-expressing MEFs harboring each of the selected siRNAs in the presence of tunicamycin. Among siRNA specific for candidate kinases, we found that siRNA specific for PERK decreased phosphorylated E3 ubiquitin ligases in LRRK (G2019S)-expressing MEFs.

PERK and kinase-active PERK-ΔN increased the levels of phosphorylated E3 ubiquitin ligases, whereas kinase-dead PERK(K618R) and siRNA targeting PERK had the opposite effect (Fig EV4A). PERK

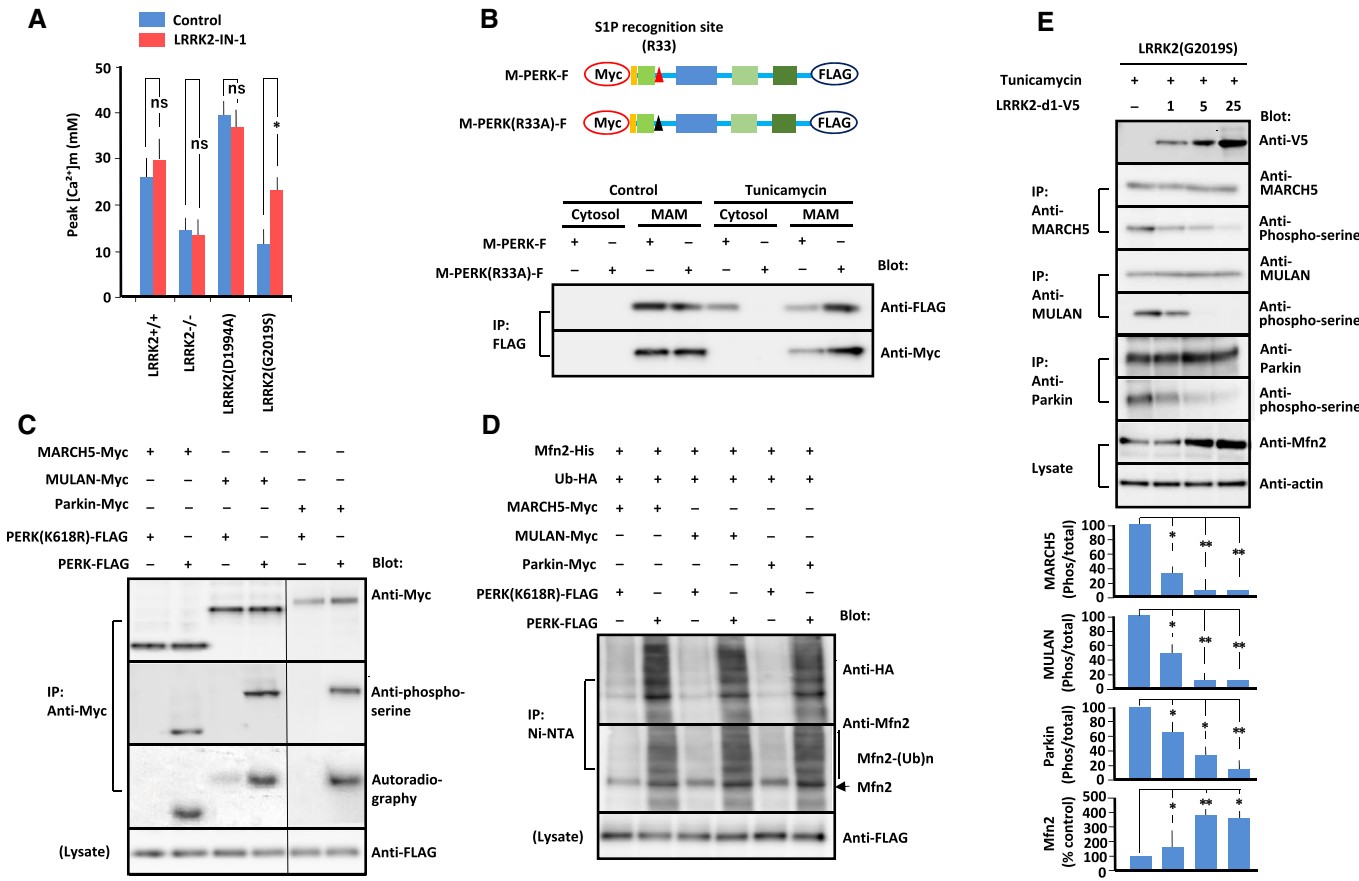

**Figure 5. PERK phosphorylates E3 ubiquitin ligases.**

A Peak values of $Ca^{2+}$ transients in MEFs of the indicated genotypes treated with LRRK2-IN-1 (1 μM). Error bars represent ± SD from six independent experiments.

B (Upper panel) Diagram showing full-length PERK tagged with Myc at the N-terminus and FLAG at the C-terminus (M-PERK-F). The S1P recognition sequence $R^{33}$SLL is mutated to $A^{33}$SLL (M-PERK(R33A)-F).
(Lower panel) Immunoblot of PERK and PERK(R33A) from transfected MEFs under tunicamycin. MAM fraction and cytosol were extracted from transfected MEFs by the Percoll gradient method. Lysates were immunoprecipitated with antibody against the FLAG epitope. Precipitated proteins were subjected to SDS/PAGE, and blots were stained with antibody as indicated to the right.

C *In vitro* kinase assay using isolated PERK, isolated E3 ubiquitin ligase, and [γ-$^{32}$P]ATP. Reaction mixture was subjected to SDS/PAGE. Blots were probed with antibody against Myc, FLAG, or phosphoserine. [γ-$^{32}$P]ATP-incorporated E3 ubiquitin ligases were visualized by autoradiography (left blot was exposed for 24 h, and right blot for 36 h).

D *In vitro* ubiquitination assay using phosphorylated E3 ubiquitin ligases, HA-tagged Ubl, and His-tagged mitofusin 2 in the presence of E1 enzyme, UbcH7, and ATP. E3 ubiquitin ligases were initially phosphorylated by PERK or PERK(K618R) *in vitro*. Phosphorylated E3 ubiquitin ligases were subjected to *in vitro* ubiquitination. Mitofusin 2 precipitated with Ni-NTA was subjected to SDS/PAGE. Blots were probed with antibody against HA or mitofusin 2. Mfn2: mitofusin 2, Ub: ubiquitin.

E Immunoprecipitation/Immunoblot of phosphorylated E3 ubiquitin ligases and mitofusin 2 in LRRK2(G2019S)-expressing MEFs transfected with the increasing amounts of LRRK2-d1-V5 (1, 5, 25 μg/10⁶ cells) treated with tunicamycin (1 μg/ml). Endogenous MARCH5, MULAN, and Parkin were immunoprecipitated with the corresponding antibody, and precipitates were immunoblotted with antibody indicated at the right. Endogenous mitofusin 2 was immunoblotted with anti-mitofusin 2 antibody. Data represent the ratio of phosphorylated to total protein levels of MARCH5, MULAN, and Parkin, and the ratio of mitofusin 2 to actin. Error bars represent ± SD from four independent experiments.

Data information: For graphs (A and E), the P values were determined by a Mann–Whitney U-test. ns = not significant, *P < 0.05, **P < 0.01.
Source data are available online for this figure.

is mainly localized at the MAM (Verfaillie *et al*, 2012). Considering the close proximity between PERK and E3 ubiquitin ligase at the MAM, it is possible that PERK directly phosphorylates E3 ubiquitin ligases under ER stress. Alternatively, PERK may be cleaved by site-1 protease (S1P), an ER-localized protease (Ye *et al*, 2000; Lichtenthaler *et al*, 2018), giving the soluble cytoplasmic domain of PERK access to E3 ubiquitin ligases. Indeed, PERK contains an RxxL motif, a known requirement for S1P processing (Espenshade *et al*, 1999; Ye *et al*, 2000), on both sites of its transmembrane domain

(Fig EV5A). Consistent with this idea, under tunicamycin treatment, we detected the cytoplasmic domain of PERK in the cytosol in MEFs transfected with PERK, but not in MEFs transfected with PERK (R33A), which lacks the putative S1P recognition site (Figs 5B and EV5B).

Next, we performed *in vitro* kinase assays using isolated PERK-ΔN or PERK(K618R) (Verfaillie *et al*, 2012), E3 ubiquitin ligase, and $^{32}$P-ATP (Fig 5C). PERK directly phosphorylated E3 ubiquitin ligase, whereas PERK(K618R) did not phosphorylate them. Previous

reports identified $S^{65}$ of Parkin as the phosphorylation site by PINK1 at the damaged mitochondrial membrane (Kondapalli *et al*, 2012; Durcan & Fon, 2015). Immunoprecipation/immunoblotting of transfected cells co-expressing Parkin or phosphorylation-defective Parkin(S65A) with PERK showed less serine-phosphorylated Parkin (S65A) than Parkin (Fig EV4B). These results implied PERK as the kinase for MARCH5, MULAN, and Parkin under ER stress and strongly suggested that PERK-mediated phosphorylation affects the activity of E3 ubiquitin ligase. Hence, we performed *in vitro* ubiquitination assays with recombinant HA-tagged Ub, E1, and E2 (UbcH7) enzymes, His$_6$-tagged mitofusin 2, and Myc-tagged E3 ubiquitin ligases, which were first *in vitro* phosphorylated by PERK or PERK(K618R), in the presence of ATP. Mitofusin 2 ubiquitinated with Ubl-HA was detected in nickel bead pull-downs in the presence of PERK, but not PERK(K618R) (Fig 5D). Thus, PERK phosphorylation increased E3 ubiquitin ligase activity.

## LRRK2 blocks PERK-mediated phosphorylation of E3 ubiquitin ligases

The final question was how LRRK2 regulates PERK-mediated phosphorylation of E3 ubiquitin ligases through its kinase activity. Immunoblotting of MEFs to detect endogenous proteins revealed similar amounts of phosphorylated PERK, but larger amounts of phosphorylated E3 ubiquitin ligases, in LRRK2(G2019S)-expressing MEFs under tunicamycin than LRRK2$^{+/+}$ MEFs (Figs 4A and EV4C). Co-expression of increasing amounts of LRRK2-D1, the binding regions of LRRK2 to E3 ubiquitin ligases (Fig 3D), in MEF expressing LRRK2(G2019S) under tunicamycin decreased PERK-mediated phosphorylation of E3 ubiquitin ligases in parallel with increased mitofusin 2 in the dose-dependent manner (Fig 5E). Based on these findings, one possible scenario is that LRRK2 directly blocks PERK-mediated phosphorylation of E3 ubiquitin ligases. To support the hypothesis that PERK-mediated phosphorylation of E3 ubiquitin ligases is dependent on the binding to LRRK2, we examined the PERK-mediated phosphorylation of E3 ubiquitin ligase in the cells co-expressing mutant LRRK2 (Fig 6A). Co-expression of LRRK2(D1994A) decreased phosphorylation of E3 ubiquitin ligases in parallel with increased binding to them, whereas co-expression of LRRK2(G2019S) increased phosphorylation of E3 ubiquitin ligases in parallel with decreased binding to them. Thus, PERK-mediated phosphorylation of E3 ubiquitin ligases was correlated with the binding of LRRK2 to E3 ubiquitin ligases, in which kinase-active LRRK2(G2019S) lost the binding thereby enhancing ligase phosphorylation and kinase-inactive LRRK2(D1994A) retained the binding thereby suppressing ligase phosphorylation.

Previous reports showed that the major substrate of LRRK2 kinase is LRRK2 itself, and that LRRK2 auto-phosphorylated at S1292 gains the ability to form a dimer, the functional unit of the enzyme (Sheng *et al*, 2012). Immunoblotting revealed that HEK293 cells transfected with LRRK2(G2019S) contained more of the phosphorylated kinase than LRRK2$^{+/+}$ or LRRK2(G1994A)-expressing cells, whereas cells expressing LRRK2(S1292A/G2019S) had lower levels of the phosphorylated kinase (Fig 6B). These results confirmed that LRRK2 auto-phosphorylates at S1292. To determine whether auto-phosphorylation affects the binding affinity of LRRK2 for E3 ubiquitin ligases, we performed immunoprecipitation/ immunoblotting of transfected cells expressing phosphomimetic

(S1292D) or phosphorylation-defective (S1292A) mutants of LRRK2 and E3 ubiquitin ligases (Fig 6C). LRRK2(S1292A) bound E3 ubiquitin ligase more strongly than LRRK2(S1292D). These results suggested that LRRK2 that is not phosphorylated at S1292 interacts with E3 ubiquitin ligases, thereby blocking PERK-mediated phosphorylation of E3 ubiquitin ligases. To confirm this idea, we examined PERK-mediated phosphorylation of E3 ubiquitin ligases with bound LRRK2(S1292A) or unbound LRRK2(S1292D) (Fig 6D). Immunoblotting showed that PERK-mediated phosphorylation of E3 ubiquitin ligases was blocked by LRRK2(S1292A), but not LRRK2 (S1292D). Thus, un-phosphorylated LRRK2 at S1292 blocked PERK phosphorylation through binding to E3 ubiquitin ligases, whereas auto-phosphorylated LRRK2 at S1292 dissociated from E3 ubiquitin ligases, allowing PERK to phosphorylate them.

Based on the proposed regulatory model of E3 ubiquitin ligases by LRRK2, the ER–mitochondrial Ca$^{2+}$ transfer in mutant LRRK2-expressing MEFs was evaluated. The ER–mitochondrial Ca$^{2+}$ transfer in LRRK2(D2019S)-expressing MEFs, the level of which was the lowest, was significantly increased by over-expression of LRRK2-D1 as well as LRRK2-IN-1 (Fig 5A, Appendix Fig S1B), where LRRK2 (D2019S), fully auto-phosphorylated, lacked the binding to E3 ubiquitin ligases (Fig 6A and B). In contrast–mitochondrial Ca$^{2+}$ transfer in LRRK2(D1994A)-expressing MEFs, the level of which was the highest, was not increased any more by two maneuvers (Fig 5A, Appendix Fig S1), where the majority of LRRK2(D1994A), not auto-phosphorylated, constitutively bound to E3 ubiquitin ligases (Fig 6A and B). In comparison with mutant LRRK2-expressing MEFs, the ER–mitochondrial Ca$^{2+}$ transfer in LRRK2$^{+/+}$ MEFs, the level of which was middle, was partially increased by two maneuvers (Figs 3C and 5A), where a fraction of LRRK2, not auto-phosphorylated, constitutively bound to E3 ubiquitin ligases (Fig 6A and B). Thus, the regulatory model of E3 ubiquitin ligases by LRRK2 could explain changes in the ER–mitochondrial Ca$^{2+}$ transfer in mutant LRRK2-expressing MEFs.

In order to overcome excess activation of E3 ubiquitin ligases and ubiquitination/degradation of MAM proteins in LRRK2 (G2019S)-expressing MEFs, it is necessary to suppress PERK activity. To this end, we treated LRRK2(G2019S)-expressing MEFs with shRNA targeting PERK (Fig 7A and B). Knockdown of PERK rescued the increased autophagic flux and increased ER–mitochondrial Ca$^{2+}$ transfer in LRRK2(G2019S)-expressing MEFs. By contrast, PERK-ΔN augmented autophagic flux and further decreased ER–mitochondrial Ca$^{2+}$ transfer in these cells. Thus, suppression of PERK kinase activity, which is required for the UPR, overcame ER–mitochondrial dysfunction in LRRK2(G2019S)-expressing MEFs.

In summary, our findings show that in addition to clearing damaged mitochondrial components as part of the canonical UPR pathway, PERK contributes to MAM formation by phosphorylating E3 ubiquitin ligases, thereby promoting the ubiquitination and degradation of substrates such as mitofusin 2. We conclude that LRRK2 is involved in the ER–mitochondrial interaction through the ubiquitination/proteasome system.

## Discussion

In this study, we demonstrated that LRRK2 binds to E3 ubiquitin ligase and blocks PERK phosphorylation and E3 ligase activity

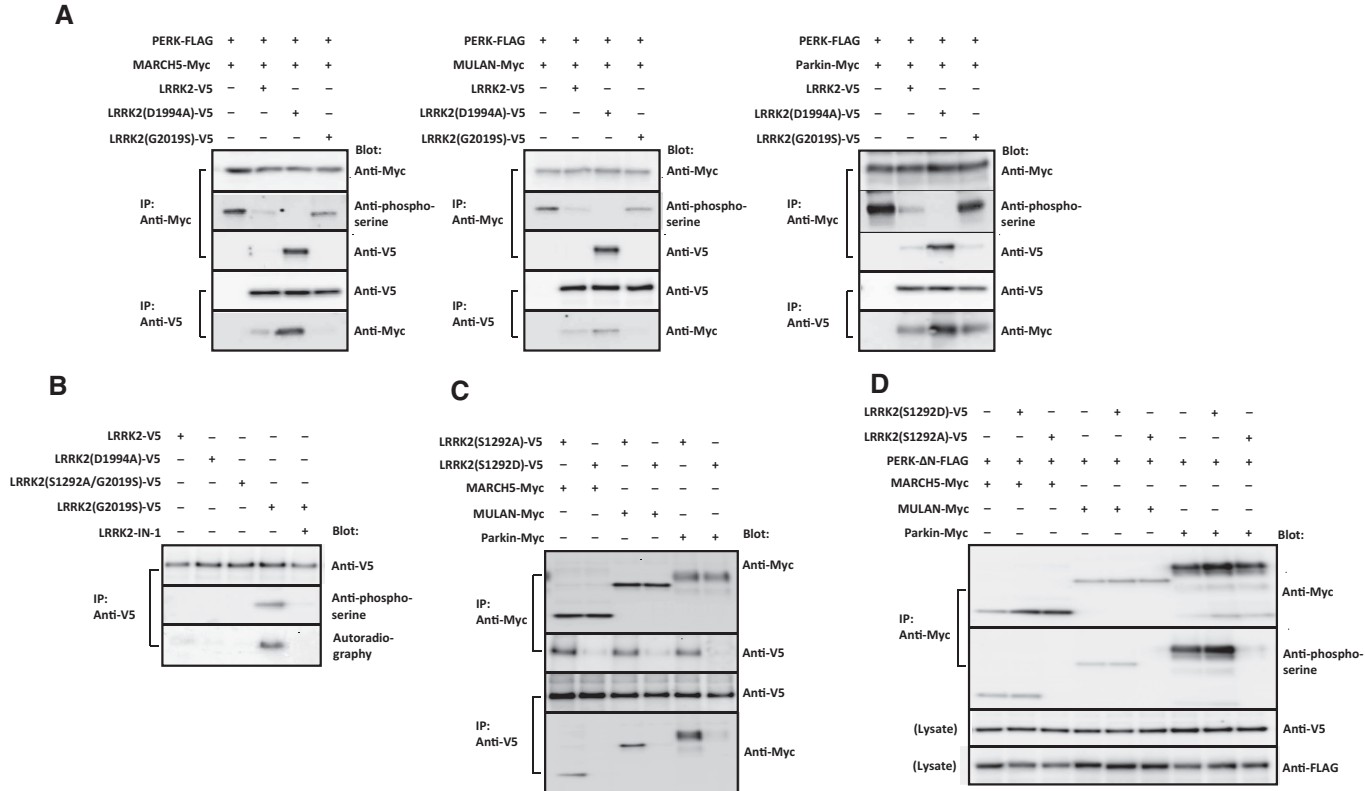

**Figure 6. LRRK2 regulates PERK phosphorylation of E3 ubiquitin ligases.**

A Immunoprecipitation/immunoblot of E3 ubiquitin ligases phosphorylated by PERK and the binding to mutant LRRK2. HEK293 cells were transfected with PERK and either MARCH5, MULAN or Parkin, along with LRRK2, LRRK2(D1994A), or LRRK2(S1292D). Cell lysates were immunoprecipitated with antibody against Myc or V5, and precipitates were immunoblotted with the antibody indicated to the right.

B In vitro kinase assay for auto-phosphorylated LRRK2 using isolated LRRK2 mutants and [γ-$^{32}$P]ATP, with or without LRRK2-IN-1 (1 μM). Mutant LRRK2 precipitated with antibody against V5 was subjected to SDS/PAGE. Blots were probed with antibody against V5 or phosphoserine. [γ-$^{32}$P]ATP-incorporated LRRK2 was visualized by autoradiography.

C Binding assays to detect interactions between phosphomimetic LRRK2(S1292D) or phosphorylation-defective LRRK2(S1292A) and E3 ubiquitin ligases. HEK293 cells were transfected with the LRRK2 mutants and E3 ubiquitin ligases, and cell lysates were immunoprecipitated with antibody against Myc or V5 epitope. Precipitated proteins were subjected to SDS/PAGE, for immunoblotting with antibody against V5 or Myc.

D Immunoprecipitation/immunoblot of E3 ubiquitin ligases phosphorylated by PERK-ΔN in the presence of LRRK2(S1292A) or LRRK2(S1292D). HEK293 cells were transfected with MARCH5, MULAN, Parkin, and PERK-ΔN along with LRRK2(S1292A) or LRRK2(S1292D). Cell lysates were immunoprecipitated with antibody against Myc, and precipitates were immunoblotted with the antibody indicated to the right.

Source data are available online for this figure.

toward MAM components. This interaction between LRRK2 and E3 ubiquitin ligases depends on the kinase activity of LRRK2; kinase-active LRRK2(G2019S) auto-phosphorylates at S1292, releasing it from E3 ubiquitin ligase, which then promotes PERK phosphorylation and ligase activation, thereby promoting ubiquitination/proteasomal degradation of MAM components. Subsequently, the reduction in the ER–mitochondrial interaction decreases IP3R/VDAC1-mediated ER–mitochondrial Ca$^{2+}$ transfer and inhibits mitochondrial energy production (Fig 7C). Of note, loss of LRRK2 causes similar phenotypic changes in the ER–mitochondrial interaction, further supporting a model in which E3 ubiquitin ligase is activated following its detachment from LRRK2.

Mitochondrial Ca$^{2+}$ is a positive effector of the tricarboxylic acid (TCA) cycle and ATP generation (McCormack et al, 1990), and also plays a major role in the regulation of autophagy and apoptosis (Cardenas et al, 2010). Efficient mitochondrial Ca$^{2+}$ uptake is

supported by the close apposition of the ER membrane and the OMM, i.e., the MAM (Gincel et al, 2001; Rapizzi et al, 2002). In LRRK2(G2019S)-expressing MEFs, mitochondrial Ca$^{2+}$ measurements using targeted recombinant Ca$^{2+}$ probes revealed reduced mitochondrial Ca$^{2+}$ uptake in response to bradykinin-mediated IP3R activation. In accordance with the Ca$^{2+}$ measurements, mitochondrial O$_2$ consumption was reduced by loss of LRRK2 or LRRK2 (G2019S), but was enhanced by LRKK2(D1994A). Therefore, kinase-active LRRK2 regulates the ER–mitochondrial interaction; kinase-dead LRRK2 activates this interaction, whereas kinase-active LRRK2 inactivates it.

We considered the possibility that LRRK2 controls the interactions between ER and mitochondrial proteins at existing sites of organelle contact. Electron microscopy and in situ PLA revealed that ER–mitochondrial contact sites were more abundant in LRRK2(D1994A)-expressing MEFs, but less abundant in LRRK2

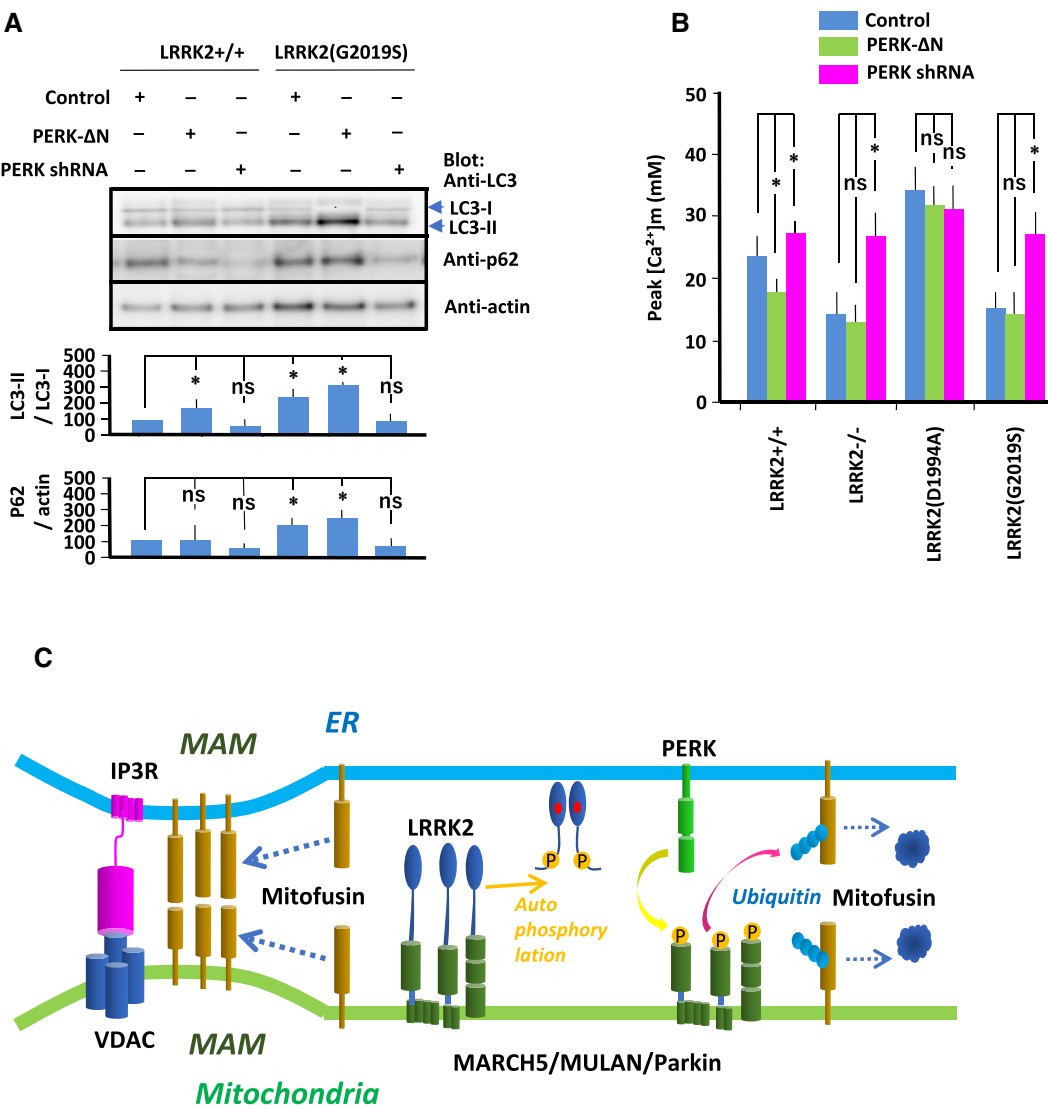

**Figure 7. Suppression of PERK rescues ER–mitochondrial dysfunction by LRRK2(G2019S).**

A   Immunoblot of LC3 and p62. MEFs of the indicated genotypes were transfected with PERK-ΔN or PERK shRNA. Endogenous LC3 and p62 levels were measured by immunoblotting. Data represent the ratios of LC3-II to LC3-I and p62 to actin, normalized against the same ratio in LRRK+/+ MEFs. Error bars represent ± SD from four independent experiments.

B   Peak values of $Ca^{2+}$ transients in MEFs of the indicated genotypes transfected with PERK-ΔN or PERK shRNA. Error bars represent ± SD from six independent experiments.

C   Schematic of the signaling pathway consisting of LRRK2, E3 ubiquitin ligase, PERK, MAM components, and the IP3R–VDAC1 complex. Un-phosphorylated LRRK2 binds to E3 ubiquitin ligases and blocks PERK phosphorylation and activation of ligase, resulting in accumulation of MAM components. Subsequently, the accumulated MAM components strengthen the ER–mitochondrial interaction, as evidenced by increased mitochondrial $Ca^{2+}$ transfer. By contrast, when LRRK2 is auto-phosphorylated, it detaches from E3 ubiquitin ligases and promotes PERK phosphorylation and activation of ligase, resulting in increased ubiquitination/proteasomal degradation of MAM components and reduced ER–mitochondrial interaction.

Data information: For graph (A), the P values were determined by a Mann–Whitney U-test. ns = not significant, *P < 0.05.
Source data are available online for this figure.

(G2019S)-expressing MEFs. Furthermore, over-expression of the synthetic tethering protein TOM-mRFP-ER rescued the decrease in mitochondrial contacts in LRRK2(G2019S)-expressing MEFs. Several different protein complexes have been proposed as ER–mitochondrial tethers, including interactions between ER-anchored IP3Rs and the mitochondrial voltage-dependent anion channel (VDAC1) mediated by GRP75 (Rapizzi et al, 2002; Szabadkai et al, 2006), homo-, and heterotypic interactions between mitochondrial mitofusin 1/2 and ER-localized mitofusin 2 (de Brito & Scorrano, 2008; Cosson et al, 2012; Filadi et al, 2015; Wang et al, 2015), interactions between the integral ER protein VAPB and mitochondrial tyrosine phosphatase-interacting protein 51 (PTPIP51) (De Vos et al, 2012; Stoica et al, 2014), interactions between Fission 1 homologue (Fis1) and ER-located Bap31 (Iwasawa et al, 2011), and interactions between

protein phosphofurin acidic cluster sorting protein 2 (PACS-2) and Bap31 (Simmen *et al*, 2005). Among the MAM components, the levels of mitofusins 1 and 2, Fos1, and PTPIP51 were decreased by LRRK2(G2019S). These results indicated that kinase-active LRRK2 directly down-regulates MAM formation by decreasing the abundance of MAM components.

Screening for the binding partners of LRRK2 revealed that this kinase functionally interacts with the mitochondrial membrane–bound E3 ubiquitin ligases MARCH5, MULAN, and Parkin through its N-terminal region. LRRK2-bound E3 ubiquitin ligases belong to the RING family (Vittal *et al*, 2015). Many E3 ubiquitin ligases exist in an auto-inhibitory state in which a region of the protein outside the catalytic domain prevents access to the active site (Deshaies & Joazeiro, 2009; Vittal *et al*, 2015). Alternatively, phosphorylation induces dimer or multi-complex formation through the RING domain, thereby activating RING E3 ubiquitin ligase (Metzger *et al*, 2014). Indeed, in the inactive state of Parkin, the RING domain is involved in the auto-inhibitory mechanism. PINK1-mediated phosphorylation of Parkin releases this autoinhibition through a conformational change (Chaugule *et al*, 2011; Shiba-Fukushima *et al*, 2012; Dove *et al*, 2015). Based on these findings, along with many reports showing that the activities of E3 ubiquitin ligases are regulated by phosphorylation (Gallagher *et al*, 2006; Smith *et al*, 2009; Lewandowski & Piwnica-Worms, 2014), we consider it likely that phosphorylated MARCH5, MULAN, and Parkin undergo conformational changes from the E2 binding–incompetent state to the E2 binding–competent state.

siRNA screening for the kinases responsible for phosphorylation of E3 ubiquitin ligases identified PERK, an ER stress sensor, as a candidate. Biochemical studies and measurements of mitochondrial $Ca^{2+}$ transfer revealed that PERK phosphorylates E3 ubiquitin ligases, thereby inducing their ligase activities toward MAM components. However, it remains unclear how ER-localized PERK can phosphorylate mitochondrially localized ligases through the gap between ER and mitochondria at the MAM (10–30 nm) (Csordas *et al*, 2006; Rowland & Voeltz, 2012). One possibility is that the distance between the ER and mitochondria at the MAM is short enough for PERK to directly phosphorylate its target ligases. Alternatively, the cytoplasmic domain of PERK (a.a. 36–114) may need to be released from the ER membrane in order to gain access to its targets. Several proteases localized at the ER membrane cleave ER-localized proteins (Ye *et al*, 2000; Lichtenthaler *et al*, 2018). Notably in this regard, ER membrane–bound ATF6, another ER stress sensor, is cleaved by S1P under ER stress, resulting in release of its cytoplasmic domain, which subsequently enters the nucleus (Ye *et al*, 2000). As with ATF6, PERK contains an RxxL motif, a known requirement for S1P processing (Espenshade *et al*, 1999; Ye *et al*, 2000), on both sites of its transmembrane domain (Fig EV5A). Under ER stress, the cytoplasmic domain of PERK was present in the cytosol, supporting the idea that the cytoplasmic domain of PERK is cleaved by ER stress, and then phosphorylates mitochondria-bound E3 ubiquitin ligases.

Beyond its canonical role as ER stress sensor, PERK is a novel mediator of ER–mitochondrial contact sites, where it phosphorylates and activates the E3 ubiquitin ligases, thereby decreasing MAM formation. Among MAM components, mitofusin 2 can tether two mitochondria together, as well as tethering mitochondria to the ER (Rowland & Voeltz, 2012). Ablation of mitofusin 2 in mouse

proopiomelanocortic neurons decreases the number of ER–mitochondrial contacts, leading to ER stress (Schneeberger *et al*, 2013). Consistent with this, loss of PINK1 or Parkin concomitant with an increase in the level of mitofusin 2 causes accumulation of misfolded proteins, leading to excess ER stress (Doyle *et al*, 2011). Thus, one interesting possibility would involve cross-talk between the ER and mitochondria in which PERK sensitizes the mitochondria to ER stress through the ubiquitination/proteasome pathway.

The final question was how LRRK2 regulates PERK-mediated phosphorylation in dependent of its kinase activity. Phosphorylation and activation of E3 ubiquitin ligases were augmented in LRRK2 (G2019S)-expressing MEFs, and these effects were suppressed by the LRRK2 kinase inhibitor LRRK2-IN-1. In line with this finding, ER–mitochondrial $Ca^{2+}$ transfer in LRRK2(G2019S)-expressing MEFs was rescued by treatment with LRRK2-IN-1. Thus, LRRK2 kinase activity was crucial for the regulation of E3 ubiquitin ligase activity. As noted above, LRRK2 auto-phosphorylates at S1292 (Sheng *et al*, 2012). Phosphorylation-defective LRRK2(S1292A) bound E3 ubiquitin ligase more strongly than phosphomimetic LRRK2(S1292D). Furthermore, LRRK2(S1292A) blocked the PERK-mediated phosphorylation of E3 ubiquitin ligases, whereas LRRK2 (S1292D) did not. Together, these results indicate that LRRK2 not phosphorylated at S1292 interacts with E3 ubiquitin ligases, thereby blocking PERK-mediated phosphorylation of E3 ubiquitin ligases. The E3 ubiquitin ligase–interacting region of LRRK2 (a.a. 1–1,515) covers an ankyrin-repeat domain, a leucine-rich repeat, a Ras complex domain, and a C-terminal Roc domain, all of which are involved in the protein–protein interaction (Gilsbach & Kortholt, 2014). S1292 is localized within the leucine-rich repeat. This finding implies that phosphorylation of S1292 induces a conformational change in the interacting sites, thereby decreasing affinity for E3 ubiquitin ligases.

In conclusion, our results show that LRRK2 regulates the phosphorylation of the mitochondrial E3 ubiquitin ligases via ER-localized PERK, thereby determining ER–mitochondrial tethering. These findings provide insight into the mechanism by which two major processes involved in PD, mitochondrial dysfunction, and ER stress, converge in modulating the PD phenotype.

## Materials and Methods

### Generation of genome-engineering MEFs

Specific targeted alterations in the *LRRK2* gene of MEFs were generated using CRISPR-Cas9 genome-engineering system (Moyer & Holland, 2015). Briefly: To create *LRRK2* knock-out MEFs, cells were electrophoretically transduced with Cas9 vector (SBI, Polo Alto, CA, USA) annealed with sgRNA; to create *LRRK2*-mutant MEFs, cells were transduced with Cas9 vector (SBI) annealed to sgRNA conjugated with homology-directed repair (HDR) dsDNA (Appendix Fig S2).

### Generation of expression vectors and site-directed mutagenesis

cDNAs encoding mouse IP3R, LRRK2, MARCH5, mitofusin 2, MULAN, MCU, MCUb, Parkin, PERK, ubiquitin, USP30, and VDAC1 were synthesized by PCR. IP3R, LRRK2, and mitofusin 2 were

ligated into pcDNA3.1/V5-His (Invitrogen/Life Technologies); MCU, MCUb, ubiquitin, USP30, and VDAC1 were ligated into pCMV-HA (Clontech Laboratories, Polo Alto, CA, USA); MARCH5, MULAN, and Parkin were ligated into pCMV-Myc (Clontech Laboratories); and PERK was ligated into 3XFLAG-CMV-13 (Sigma-Aldrich, St Louis, MO, USA). Deletion constructs of LRRK2 [LRRK2-d1 (a.a. 1–1,515), LRRK1-d2 (a.a. 1,516–2,527)] were synthesized by PCR and ligated into pcDNA3.1/V5-His. A deletion construct of PERK [PERK-ΔN (a.a. 10–1,114)] was synthesized by PCR and ligated into 3XFLAG-CMV-13. Mutations of $D^{1994}$ to A and $G^{2019}$ to S in LRRK2 were introduced to create kinase-dead and kinase-active forms of the protein, respectively. Mutations of $S^{1292}$ to A and $S^{1292}$ to D in LRRK2 were introduced to create phosphorylation-defective and phosphomimetic LRRK2, respectively. Mutations of $H^{43}$ to W in MARCH5, $C^{339}$ to A in MULAN, and $C^{431}$ to A in Parkin were introduced to create ligase-inactive MARCH5(H43W), MULAN(C339A), and Parkin(C431A), respectively. Mutation of $W^{403}$ to A in Parkin was introduced to create ligase-active Parkin(W403A). Mutation of $C^{77}$ to S in USP30 was introduced to create deubiquitinase-inactive USP30(C77S). Mutation of $K^{618}$ to R in PERK was introduced to create a kinase-defective form of the protein, and mutation of $R^{33}$ to A was introduced to create an S1P-resistant form. Mutations were created using the QuikChange site-directed mutagenesis kit (Stratagene, La Jolla, CA, USA).

To construct the ER–mitochondrial linker (TOM-mRFP-ER), mRFP was targeted to the ER using the C-terminal ER localization sequence of the yeast UBC6 protein (residues 233–250 MVYI-GIAIFLFVGLFMK). This construct was complemented with the N-terminal mitochondrial localization sequence of mouse TOM70 (residues 1–63; Kornmann, 2013).

### RNAi

shRNA oligonucleotides specific for the target sequence of mouse IP3R, PERK, or VDAC1 were designed (Appendix Fig S3A) and ligated into expression vector pcDNA6.2-GW-miR (Clontech Laboratories). Cultured cells were transfected with shRNA vectors using Nucleofector™ technology (Amaxa Biosystems, Cologne, Germany). Efficiency of knockdown was verified by Western blot (Appendix Fig S3B). Based on the inhibitory effect of endogenous protein expression, one of three candidate shRNAs was chosen and used in subsequent experiments.

### Reagents

Reagents and kits were obtained from the indicated suppliers: ADP-Glo Assay kit (Promega, Madison, WI, USA), alkaline phosphatase (CIAP) (TAKARA, Kusatsu, Shiga, Japan), ATP Determination Assay kit (Molecular Probes, Eugene, OR, USA), 2APB (Sigma-Aldrich), bafilomycin A1 (Sigma-Aldrich), BD Matchmaker Pretransformed cDNA Library (Clontech Laboratories), bradykinin (Sigma-Aldrich), Citrate Synthase Assay kit (Sigma-Aldrich), Duolink *in situ* assay (Sigma-Aldrich), E1 enzyme (Abcam, Cambridge, UK), anti-FLAG Sepharose (Sigma-Aldrich), and FLAG peptides (Sigma-Aldrich), Lipofectamine 3000 (Thermo Fisher, Waltham, MA USA), LRRK2-IN-1 (Cayman Chemical, Ann Arbor, MI, USA), MISSION siRNA mouse kinase panel library (Sigma-Aldrich), Mitotracker (Molecular Probe), Ni-NTA beads (Qiagen, Germantown, MD, USA), Ru360

(Sigma-Aldrich), and tunicamycin (Sigma-Aldrich), UbcH7 (Abcam), and Vybrant MTT Cell Proliferation Assay Kit (Thermo Fisher).

### Antibodies, immunoprecipitation, and immunoblotting

Antibodies were obtained from the indicated suppliers (Appendix Fig S4): anti-actin (Abcam), anti-Bap31 (Abcam), anti-calnexin (Abcam), anti-DRP1 (Abcam), anti-Fis1 (Merck Millipore), anti-GRP75 (Abcam), anti-IP3R (Abcam), anti-LC3 (MBL, Nagoya, Aichi 460-0008, Japan, Merck Millipore, Burlington, MA, USA), anti-LRRK2 (Abcam), anti-MARCH5 (Abcam), anti-mitofusin 1 (Abcam), anti-mitofusin 2 (Abcam), anti-MULAN (United State Biological), anti-Parkin (Abcam), anti-PTPIP51 (Abcam), anti-p62 (MBL), anti-PERK (Abcam), anti-VAPB (Abcam), anti-VDAC1(Abcam), anti-FLAG (Sigma-Aldrich), anti-HA (MBL), anti-Myc (MBL), anti-phosphoserine (Abcam), and anti-V5 (Thermo Fisher). Transfected cells were incubated for 2 days, harvested, and then lysed in lysis buffer (50 mM Tris–HCl [pH 8.0], 150 mM NaCl, and 0.1% SDS) for immunoblotting or in TNE buffer (50 mM Tris–HCl, pH 7.4 150 mM NaCl, 5 mM EDTA, 1% NP-40, 0.25% Na-deoxcholate, and 1 mM NaF) for immunoprecipitation/immunoblotting. Immunoprecipitation/immunoblot analyses were performed using standard protocols. The specificity of anti-phospho-serine antibody to phosphorylated serine was confirmed by immunoblotting of E3 ubiquitin ligases extracted from transfected cells treated with or without phosphatase (Appendix Fig S5A).

### Subcellular fractionation

MAM, mitochondria, and microsomes were isolated from cells using Percoll gradient fractionation (Bozidis *et al*, 2007). In brief, cells were harvested and lysed in Sucrose Homogenization Medium (0.25 M sucrose, 10 mM HEPES, pH 7.4). Differential centrifugation was used to isolate the post-nuclear supernatant from nuclei and cellular debris. The total microsomal fraction and crude mitochondrial fraction were isolated by centrifugation at 10,300 *g*. Following this step, the microsomal fraction was isolated following centrifugation at 100,000 *g*. The resulting supernatant from this spin is the "cytosol" fraction, and it was concentrated by using Amicon Ultra 15-ml filters. The crude mitochondrial fraction was purified through a self-generating Percoll gradient, and the collected mitochondrial and MAM fractions were further purified by centrifugation at 6,300 *g*. The MAM was then collected following centrifugation at 100,000 *g*. Equivalent amounts of protein from each fraction were analyzed by SDS–PAGE and immunoblot.

### Oxygen consumption

Oxygen consumption rate (OCR) was measured at 37°C using an XF96 Extracellular Flux Analyzer (Seahorse Bioscience, North Billerica, MA). Cells seeded into 96-well plates ($1 \times 10^4$ per well) were loaded into the machine for determination of oxygen concentration. Cells were exposed to oligomycin (1 μM), carbonyl cyanide p-trifluoromethoxyphenylhydrazone (FCCP; 300 nM), and rotenone (100 nM) plus actinomycin (100 nM). After each injection, OCR was measured for 5 min. Representative traces are

shown in Fig 1C. Every point represents the average of four different wells. Basal OCR was calculated as the difference between OCR measurements taken before and after oligomycin. Maximum OCR was calculated as the difference between the OCR measurements taken after FCCP and after exposure to rotenone plus actinomycin.

## ATP production

ATP content in MEFs was measured using the ATP Determination Assay kit (Molecular Probes). ATP concentration was calculated using an ATP standard curve.

## Calcium imaging

Plasmids for expression of the mitochondrial $Ca^{2+}$ sensor (pcDNA3.0-2mt-cameleon) and the ER $Ca^{2+}$ sensor (D1ER pcDNA3) were kindly provided by Dr. Roger Tsien (University of California; Palmer *et al*, 2006).

To measure the $Ca^{2+}$ concentration in mitochondria and ER, MEFs plated on 3.5-cm confocal dishes were transfected with pcDNA3.0-2mt-cameleon and D1ER pcDNA3 using Lipofectamine 2000 (Invitrogen, Carlsbad, CA, USA). Along with the $Ca^{2+}$ sensor expression plasmids, an expression vector containing the target construct was transfected (1:10 molar ratio of $Ca^{2+}$ sensor expression plasmid to target expression vector). After 2 days, cells were rinsed twice and then maintained in Hanks' balanced salt solution (HBSS: 142 mM NaCl, 5.6 mM KCl, 1 mM $MgCl_2$, 2 mM $CaCl_2$, 0.34 mM $Na_2HPO_4$, 0.44 mM $KH_2PO_4$, 4.2 mM $NaHCO_3$, 10 mM HEPES, and 5.6 mM glucose [pH 7.4]). Calcium imaging experiments were performed using an LSM 700 microscope (Carl Zeiss, Oberkochen, Germany). Dual-emission ratio imaging of cameleon was accomplished using the BP420/10 excitation filter, a 440/520 dichroic mirror, and two emission filters (BP472/30 for cyan fluorescent protein and BP542/27 for YFP), which were alternated using a filter changer. Exposure time was 100 ms, and images were collected every 3 s. Baseline (50-s) measurements were acquired before the first pulse of bradykinin (BK). BK was dissolved in HBSS, and the working concentration was 2.5 μM. In some experiments, MEFs were preincubated with 2-amino-ethoxydiphenyl borate (2-APB, 20 μM) for 30 min at room temperature prior to stimulation with BK. The free $Ca^{2+}$ concentration in mitochondria or ER was determined as previously described (Palmer *et al*, 2006). Peak values of $Ca^{2+}$ transients in MEFs transfected with constructs or treated with agents were compared with those in MEFs transfected with empty vectors or vehicles, which had no significant effects on these values (Appendix Fig S5B).

## *In situ* proximity ligation assay

Quantification of protein interactions (< 40 nm) as individual fluorescent dots was performed using the Duolink *in situ* assay (Sigma-Aldrich). MEFs on slides were fixed and permeabilized. The samples were probed with rabbit anti-IP3R antibody and mouse anti-VDAC1 antibody, and then with anti-mouse IgG and anti-rabbit IgG conjugated to oligonucleotide extensions. In this system, if the oligonucleotides are within a distance of 40 nm, they hybridize with subsequently add connector oligonucleotides to form a circular DNA

template, which is ligated and subsequently amplified to create a single-stranded DNA product. In MEFs, the size of ER–mitochondrial junctions (10–25 nm) enabled proximity ligation and subsequent detection by hybridization of Texas red-labeled oligonucleotide probes. Fluorescence was analyzed on a Zeiss inverted fluorescence microscope. Each fluorescent dot represents the formation of one IP3R–VDAC1 interaction.

## Cell viability assay

Cell viability was measured using the Vybrant MTT Cell Proliferation Assay Kit (Thermo Fisher). MEFs were incubated with MTT solution in phenol red-free DMEM for 4 h at 37°C. The MTT assay involves the conversion of water-soluble MTT (3-[4,5-dimethylthiazol-2-yl]-2,5-diphenyltetrazolium bromide) to insoluble formazan. The formazan is then solubilized by DMSO, and the concentration is determined by measuring optical density at 570 nm.

## Citrate synthase assay

Citrate synthase, the initial enzyme of the tricarboxylic acid (TCA) cycle and an exclusive marker of the mitochondrial matrix, was measured using the Citrate Synthase Assay kit (Sigma-Aldrich). Whole-cell lysates of MEFs were incubated with acetyl co-enzyme A (acetyl CoA) and oxaloacetic acid, yielding CoA with a free thiol group. The CoA then reacted with 5,5′-dithiobis-(2-nitrobenzoic acid) (DTNB) to form yellow TNB, which was spectrophotometrically measured at 412 nm.

## *In vitro* kinase assay

Reactions were performed at 30°C for 30 min in 40 μl of kinase buffer (20 mM HEPES, pH 7.5, 50 mM KCl, 1.5 mM DTT, 2 mM $MgCl_2$, 0.1 mM ATP) containing 6 μCi of $[\gamma\text{-}^{32}P]ATP$ (1 Ci = 37 GBq), 2 μg of E3 ubiquitin ligase isolated from cell lysates with anti-Myc Sepharose, and 4 μg of PERK isolated from cell lysates with anti-FLAG Sepharose and FLAG peptide (Sigma-Aldrich). Reaction mixtures were subjected to 12 % SDS/PAGE and visualized by autoradiography on a phosphorimager.

## *In vitro* ubiquitination assay

Reactions were carried out in a total volume of 100 μl of ubiquitination buffer (50 mM Tris–HCl, pH 7.5, 5 mM $MgCl_2$, 1 mM DTT, 100 mM NaCl) containing 90 nM E1 enzyme, 4 mM ATP, 0.4 mM HA-tagged Ub, 4 μg UbcH7, 4 μg His-tagged mitofusin 2, and 2 μg of MARCH5, MULAN, or Parkin. The latter proteins were first phosphorylated by PERK or PERK(K618R) in kinase buffer at 37°C for 30 min and then isolated with anti-Myc Sepharose. The reaction was incubated for 2 h at 37°C, followed by centrifugation at 3,000 *g*. The pellet containing His-tagged mitofusin 2 was washed with ubiquitination buffer and subjected to SDS/PAGE. The supernatant was incubated in Ni-NTA binding buffer (50 mM $NaH_2PO_4$, 300 mM NaCl, 10 mM imidazole, pH 8.0) containing 20 ml of Ni-NTA beads. The Ni-NTA beads were washed with Ni-NTA binding buffer and then subjected to SDS/PAGE.

## Kinase activity assay

LRRK2 kinase activity was measured using the ADP-Glo Assay kit. The kinase reaction was performed with LRRK2 isolated from MEFs with anti-LRRK2 antibody conjugated to Protein G Sepharose, LRRKtide (RLGRDKYKTLRQIRQ) as a substrate, and ATP. ADP formed from the kinase reaction was converted into ATP, and the newly synthesized ATP was measured using a coupled luciferase/luciferin reaction.

## Yeast two-hybrid library screening

The BD Matchmaker Pretransformed cDNA Library was used in this study. Yeast strain AH109 was transformed with the bait plasmid pGBDU-C1 encoding the N-terminus (aa. 1–1,515) of LRRK2 and then screened against a mouse brain cDNA Matchmaker library (BD Biosciences, San Jose, CA, USA). Interacting proteins were identified by plasmid sequencing and BLAST searching. To confirm the interaction with the identified prey DNA, the indicated regions of LRRK2 and prey DNA were cloned into the GAL4-DNA binding-domain and GAL4-DNA activation-domain plasmids, respectively. The resultant plasmids were transformed into yeast strain PJ69-4A, and the interaction between the two proteins was tested in the yeast two-hybrid system. Interactions between binding- and activation-domain fusion proteins were scored based on yeast growth.

## Electron microscopy

MEFs were incubated in 0.1 M phosphate buffer containing 2.5% glutaraldehyde and then post-fixed in 1% osmium tetroxide. The cells were embedded in Spurr's resin. Embedded samples were cut into ultrathin sections. Sections were counterstained with 2% uranyl acetate and lead citrate. Micrographs were obtained at 5,000× or 12,000× magnification. To obtain the interface percentage, mitochondrial perimeter and area, and the lengths of ER interfaces, were measured using Metamorph. Distances of ER–mitochondrial interfaces were determined by measuring the shortest distance between the ER membrane and the mitochondrial OMM at two sites for each contact. Eight images, each of which contained 2–4 mitochondria, were obtained from 5 MEFs of indicated genotypes.

## Immunohistochemistry

MEFs were grown on sterile glass coverslips and fixing with 4% PFA in PBS for 10 min at room temperature. After blocking with PBS containing 1% BSA and permeabilizing with 0.1% Triton X-100, cells were then stained with antibody against target protein diluted according to the manufacturer's recommendation in blocking buffer overnight at 4°C. Cells were washed with PBS and stained with secondary antibody (1:500 in blocking buffer) for 2 h at room temperature. Samples were mounted with ProLong Gold Antifade (P-36930, Life Technologies), and randomly chosen field images were obtained in an LSM 700 microscope (Carl Zeiss, Oberkochen, Germany).

## Mitochondrial morphology

MEFs were grown on sterile glass coverslips and stained with Mito-tracker (green) and DAPI (blue) for 2 h. For analysis, we used the Mitochondrial Network Analysis (MiNa) toolset, a combination of different ImageJ macros that allows the semiautomated analysis of mitochondrial networks in cultured mammalian cells. Briefly, the image was converted to binary by thresholding following the conversion to a skeleton that represents the features in the original image using a wireframe of lines of one pixel wide. All pixels within a skeleton were then grouped into three categories: end point pixels, slab pixels, and junction pixels. The parameters used in the study were (i) individuals, punctate, rods, and large/round mitochondrial structures; (ii) networks, mitochondrial structures with at least a single node and three branches; (iii) the mean number of branches per network; and (iv) the average of length of rods/branches. Ten randomly chosen fields containing between 10 and 15 cells were used to quantify the pattern of mitochondria. We classify the mitochondrial morphology into three different subtypes according to the length of the branches: filamentous (long and spaghetti-like shape; branch > 2.3 μm), fragmented (completely dotted; branch < 1.8 μm), and intermediate pattern (when both filamentous and fragmented mitochondria were found; 1.8 μm ≥ branch ≥ 2.3 μm).

## siRNA library screening

The MISSION siRNA mouse kinase panel library was used in this study. A pool of three siRNA molecules targeting the same gene was generated by mixing equal amounts of the three individual siRNA oligonucleotides that were used for screening. Briefly, 20,000 cells were seeded into 96 well microplates pretreated with Lipofectamine 3000 (Thermo Fisher Scientific) and allowed to incorporate siRNA overnight. The medium was replaced the following day, and the cells were cultured for 48 h after transfection. Cell viability was measured using the Vybrant MTT Cell Proliferation Assay Kit.

The Z-score was calculated to assess changes in cell viability in a single well relative to the mean cell viability across all wells on the same plate. Levels of apoptosis were measured described as above for each 96-well plate and represented as C = [C1, C2, …..C96]. The score for the number of apoptotic cells in a well was $Z_i = (C_i - \mu)/\sigma$, where μ and σ are the mean and standard deviation of C, respectively.

## Statistical analysis

All statistical analyses were performed using GraphPad Prism. All samples were first subjected to a D'Agostino–Pearson omnibus normality test. If values were distributed in a Gaussian manner, *t*-test was used for paired comparisons, and one-way ANOVA followed by Bonferroni's multiple comparison tests for multiple comparisons. For non-Gaussian distributions, a Mann–Whitney *U*-test was used for paired comparisons, and a Kruskal–Wallis nonparametric ANOVA test was used for multiple comparisons.

**Expanded View** for this article is available online.

## Acknowledgements

We thank Dr. Roger Y Tsien for plasmid mt-cameleon and D1ER pcDNA3. This study was supported by research grants from the Ministry of Education, Culture, Sports, Science and Technology of Japan (T.T. and A.K.) and Center of Innovation program (COI-STREAM) grants from the Ministry of Education, Culture, Sports, Science and Technology of Japan.

## Author contributions

TT, YO, TI, and SS carried out most of the experiments. TT wrote the manuscript. AK provided advice on project planning and data interpretation.

## Conflict of interest

The authors declare that they have no conflict of interest.

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
