## [Review Process File · The EMBO Journal]

LRRK2 regulates endoplasmic reticulum–mitochondrial tethering through the PERK-mediated ubiquitination pathway

Toshihiko Toyofuku, Yuki Okamoto, Takako Ishikawa, Shigemi Sasawatari, and Atsushi Kumanogoh

Review timeline:

Submission date:	9th Oct 2018
Editorial Decision:	4th Dec 2018
Revision received:	25th Mar 2019
Editorial Decision:	27th Jun 2019
Revision received:	30th Jul 2019
Editorial Decision:	1st Oct 2019
Revision received:	24th Oct 19
Accepted:	5th Nov 2019

Editor: Daniel Klimmeck

Transaction Report:

1st Editorial Decision

4th Dec 2018

Thank you for submitting your manuscript (EMBOJ-2018-100875) for consideration by The EMBO Journal. We have now received reports from three referees, which I enclose below. In light of these comments, I am afraid we decided that we cannot offer publication in The EMBO Journal.

As you can see, the referees appreciate that the analysis extends previous work. However they also raise major concerns with the analysis that I am afraid preclude publication here. In more detail, the referees consistently state major issues with unsupported claims and inconsistencies in the proposed model. Further they point to the insufficiently addressed endogenous relevance of the findings as well as concerns regarding the robustness of the results presented.

Given these negative opinions from good experts on the field, and that we need strong support from the referees to move on, I am afraid we cannot offer to publish your study in The EMBO Journal.

Thank you in any case for the opportunity to consider this manuscript. I regret we cannot be more positive on this occasion, but we hope nevertheless that you will find our referees' comments helpful.

REFeree REPORTS:

Referee #1:

This study investigates known aspects of ER-mito tethering. The novelty here is to introduce a novel regulatory mechanism via LRRK2. Evidence is provided that suggests LRRK2 binding to mitochondrial ubiquitin ligases affecting catalytic activity, which however must be proven in

appropriate enzyme assays. From the perspective of LRRK2 as one of the most important Parkinson's disease gene products, the impact on ER-mito tethering protein turnover offers new insight into PD pathogenesis.

Having said that, I have strong doubts about the experimental system and some interpretations. I find it incomprehensible why LRRK2 knockout and LRRK2 hyperactive MEF cells have the same effects. That makes absolutely no sense, at least not in the straightforward model the authors put forward. LRRK2^{-/-} should have the strongest protective effect, similar to the inactive (maybe even dominant-negative) D1994A mutant, whereas conversely G2019S should have a slightly stronger disturbing effect than WT. That would be the correlation with known LRRK2 kinase activities. Note that G2019 is estimated to have only some 2-fold elevated kinase activity. This issue has got to be convincingly resolved. A major deficiency is that at no point expression levels and kinase activities are measured. All effects could simply be due to different expression levels of the LRRK2 proteins.

Along similar lines (lack of clear correlation with LRRK2 kinase activities), note that the d1 deletion construct lacking the LRRK2 kinase (and WD40) domain, cannot auto-phosphorylate itself. Therefore, the authors' claims about auto-phosphorylation of the LRRK2 N-terminal portion are not entirely supported by the data. There are possible explanations for this, but need to be included in a convincing discussion.

An issue is also that all analyses were done with transfected, (over-)expressed proteins. Efforts must be undertaken to confirm key findings at the level of the endogenous proteins.

Finally, I don't get the point of Fig. 5. The authors show phosphorylation of the ubiquitin ligases correlated with E3 activity. The authors go on to say that direct LRRK2 kinase activity towards these ubiquitin ligases could not be detected, and drop the point of Fig. 5. Instead, they move on to claim that auto-phosphorylated LRRK2 N-terminal domains could de-repress these ubiquitin ligases. For the ubiquitin ligase parkin, phosphorylation at S65 is an established mechanism. Allosteric regulation by an activator protein (LRRK2) would be a novel mechanism that must be unequivocally proven in enzyme activity assays.

Detail criticisms:

- 1) Specify VDAC1 (instead of VDAC) throughout the text.
- 2) Page 12, line 10: [stress] "decreased cell viability in LRRK2^{-/-} and LRRK2(G2019S)-expressing MEFs" MORE STRONGLY not less.
- 3) Page 14, line 15: LRRK2 not LRRK1
- 4) Comparisons in Fig. 5A,5B,5C,6A must be done on the same blot!
- 5) Page 19, line 18: [IP/blots for S1292D and S1292A] Figure 6C not 6B
- 6) Page 23, line 7: "levels of mitofusins, Fis1 (not Fos1) and PTPIP51 were decreased by kinase-active LRRK2" is not quite, all these proteins were decreased in LRRK2^{-/-} compared to LRRK2^{+/+}.
- 7) Moreover, I don't find the contrasting expression changes for DRP1 and Fis1 well explained. How would such differential expression of mitochondrial fission factors affect ER-mito tethering, exactly?
- 8) Very last sentence: "It remains unclear how E3 ubiquitin ligases are phosphorylated." Yes, indeed. Efforts should be undertaken to resolve this issue (see above).

Referee #2:

This manuscript entitled "LRRK2 regulates ER-mitochondrial tethering and mitochondrial energetics" describes the role of LRRK2 in mitochondria energetics by using CRISPR/Cas9 modified MEF cells, and links this function to a modification of ER-mitochondria contacts.

Although some results are interesting, the data presented need to be confirmed with appropriate controls, and with complimentary experiments.

Major points:

1. The characterization of the cell lines used in this manuscript is not fully described. The authors must show the molecular analysis of their clones (Sanger sequencing of the alleles allowing to verify that the genetic modification is homozygous), a Western blot analysis to verify that no protein is present in the KO cell line, and that LRRK2 is still expressed and at a similar level in the knock-in cell lines. Besides, the increased kinase activity of the G2019S could be tested (using model substrate such as LRRKtide or Nictide, for instance).
2. Many important controls are missing in this study:
 - a. The experiments seem to have been performed in only one CRISPR/Cas9 edited clone. How can the authors rule out that the phenotypes they observe do not result from a clonal effect (and thus not from the mutation they introduced)? Importantly, rescue experiments have been performed for the knock-out cell line (Figure 4B), but this was only analyzed for one phenotype. For the knock-in lines (for which no rescue can be performed), the most important experiments should be repeated in several MEF clones.
 - b. Figure 2C, D, F: these experiments include overexpression, silencing (with shRNAs), and treatment with drugs: only one control is shown (which is not described); each of these experiments must have their own control: empty vector (for OE), control shRNA (for silencing), vehicle (for drug treatment). Besides, 1 shRNA per gene is not enough: the standard is to use at least two distinct shRNAs.
 - 3) The silencing (IP3R, VDAC) and the OE (IP3R, VDAC, MARCH, Parkin etc...) efficiency have to be controlled by WB.
 - c. The specificity of the anti-phosphoserine antibody has to be tested, for instance by treating the WB membrane with a phosphatase (CIP) before the incubation with the anti-phosphoserine antibody.
 - d. Figure 2E: the bafilomycin-treated samples cannot be on the same WB as non-treated samples. They must have their own loading control. Besides, p62 level in these samples should be shown.
 - e. Controls need to be added to the PLA experiments. Are the anti-IP3R and VDAC antibodies able to label the endogenous proteins by immunofluorescence? Does the number of PLA dots detected depend on the expression of IP3R and VDAC (perform the experiment in cells silenced for IP3R or VDAC)?
3. The data concerning ER-mitochondria tethering seem to be over-interpreted to this reviewer. Indeed, the PLA experiment is not sufficient to draw the conclusion that ER-mitochondria tethering is affected in the different cell lines. Indeed, this experiment just shows that there are slightly less IP3R-VDAC complexes in LRRK2^{-/-} (and G2019S) cells compared to controls (and slightly more in D1994A mutants compared to controls). Most importantly, PLA results are correlated to the levels of the two targets used; it seems that the level of VDAC is reduced in LRRK2^{-/-} cells compared to LRRK2^{+/+} cells (Fig 3B).

If the authors want to show that the tethering is modified, they need to show it properly. The best way to test the tethering is to perform electron microscopy and to quantify ER-mitochondria contacts in the different cell lines.
4. Figure 3B: the level of the different proteins must be quantified (ideally from several MEF clones).

Minor points:

1. Many typos are present on the figures. For instance, Fig3B Mifn2, Fig1E, ECOR etc...
2. English needs to be corrected.
3. FigS1: the method used to quantify mitochondria fragmentation is not described. These data do not include statistics.

Referee #3:

This manuscript investigates the role of LRRK2 in the regulation of the ER-mitochondria interface in genome-edited mouse embryonic fibroblasts knocked out for LRRK2, or expressing the common kinase hyperactive LRRK2 G2019 mutation or a kinase dead variant (D1994A). The data show a common phenotype for LRRK2^{-/-} and LRRK2 G2019 cells, associated with decreased mitochondrial respiration and ATP synthesis, and indicative of increased mitochondrial damage and autophagy, decreased ER-to-mitochondria calcium-transfers and ER-mitochondrial tethering, and

increased vulnerability to ER stress. Opposite phenotypes are observed for the D1994A variant. In an attempt to investigate the mechanisms underlying regulation of the ER-mitochondria interface, the authors evaluated the physical/functional interaction between LRRK2 variants and three E3 ubiquitin ligases reported to regulate the ER-mitochondria tethering protein Mitofusin-2. Their data suggest that LRRK2 kinase hyperactivity overactivates these ligases, leading to enhanced proteasomal degradation of Mfn2 and dissociation of ER and mitochondria.

Despite the potential interest of this study, experimental flaws complicate the interpretation of the data under their present form. The most surprising finding is the observation of identical phenotypes in cells knocked out for LRRK2 or expressing the overactive G2019S variant, when one would have rather expected the kinase inactive D1994A to result in effect similar to total LRRK2 loss.

Specific comments

1) Evidence for altered ER-mitochondria tethering is mostly indirect. The conclusions would be strengthened by a thorough electron microscopy analysis of this interface in the different cell lines. Western blot analysis of proteins of the MAM fractions in 3B should be quantified for reliable conclusions to be drawn. Drp1 levels seem to follow a pattern opposite to that of Mfn1/2, how do the authors interpret this finding? Does expression of the TOM-mRFP-ER tether also "rescue" the LRRK2 G2019S and +/- ER-mitochondria tethering defect evaluated by proximity ligation assay in Figure 3A?

2) To draw reliable conclusions about the impact of LRRK2 on intracellular calcium homeostasis, investigation of ER-mitochondria calcium transfers should be accompanied by parallel analyses of the intracellular calcium pools (are ER calcium pools unchanged in cells expressing the different LRRK2 variants?) and ER-cytosol calcium fluxes.

3) The analysis of the effect of LRRK2 domains on the regulation of ER-mitochondria calcium transfer in a LRRK2-deficient background is absolutely unclear (Figure 4A-B). For any conclusion to be drawn, the study should be repeated with the G2012S and D1994A variants. One would expect the normal LRRK2 protein to have an effect similar to that of the G2012S variant, i.e. reduction in ER-mitochondria calcium transfer. As LRRK2 G2012S was shown to decrease ER-mitochondria calcium transfer to an extent similar to that of LRRK2-deficiency, why should kinase-active LRRK2 enhance calcium transfers in the absence of LRRK2? Effect of individual E3 ubiquitin ligases should be explored on ER-mitochondria calcium transfers in Figure 4F.

4) To clarify the effect of the LRRK2 kinase domain on activation of E3 ligases and ubiquitylation of Mfn2 (Figure 6A), LRRK2-In-1 should also be investigated in cells expressing the inactive LRRK2 D1994A variant and in "wild-type" cells. Would we not expect LRRK2-In-1 to shift the situation towards that in cells expressing the inactive variant rather than to that in wild-type cells?

5) It is unclear how autophagy flux was analysed (Figure 2E): it is noted that values were normalized against controls, but what is meant by "control" here? How did the authors take into account the effect of bafilomycin? Why don't they analyse LC3II/I ratios? Why did they not analyse quantitatively p62 levels to show that they indeed "followed a similar pattern"?

Additional comments:

- The authors should indicate how they validated the expected gene editing events in MEFs and show the corresponding results. How did they ensure that both LRRK2 alleles were modified in cells? How efficient was the approach?
- Quantitative data originating from several independent experiments should be provided for each western blot analysis
- Effects of siRNAs (VDAC/IPR3...) and protein overexpression on the specific target transcript/protein should be systematically shown (by RT-PCR/western blot and/or immunocytochemistry)
- Precise references should be provided for the antibodies used
- The origin of the LRRK2 expression vector should be indicated where appropriate in the Experimental methods
- In Figure 2F, "thapsigargin" should be "thapsigargin"
- The present version was extremely difficult to read because the authors forgot to accept the English corrections throughout the text.

Thank you for submitting the revised version of your manuscript and my apologies for the unusual delay in getting back to you due to internal detailed discussions. We have sent your revised manuscript back to the three original referees for re-evaluation, and we have received feedback from all of them, which I enclose below.

As you will see, all referees state that the manuscript has been largely improved and are supportive, however both referees #2 and referee #3 state remaining concerns that have to be conclusively addressed to support publication. We have re-discussed those points in detail in the team and decided to invite you for an additional final revision of the study considering the following points:

>> Improve EM data quality and provide quantification of the findings (ref#2, pts.1-3).

>> Consolidate you results by additional assays testing differential ligase binding of the WT vs mutant LRRK2 (ref#3).

>> Address additional referee points regarding textual points, additional controls required and data representation (all referees).

Please revise the manuscript to see if you can adjust claims made or introduce caveats where appropriate. Note that once above points are solved, we are happy to swiftly move on with acceptance and publication of the work.

 REFEREE REPORTS:

Referee #1:

The manuscript has greatly improved by the identification of PERK as the kinase activating LRRK2-autophosphorylation liberated ubiquitin ligases. Now the findings and conclusions make much more sense overall. Please consider a few details:

1) The authors determine global serine phosphorylations of the ubiquitin ligases. At least for parkin, a stimulatory phosphorylation is known to occur at serine-65. Is this the residue targeted by PERK?

2) page 3, line 8: correct domain names Ras of complex proteins (Roc) and C-terminal of Roc (COR) [Bosgraaf & Haastert (2003) BBA 1643:5]

3) double-check text annotation for Fig. S1D

4) D1994A behaves similar or even stronger than +/+ in contrast to -/- and G2019S in most of the experiments. However, LRRK2(D1994A) cells show the same electron-dense material as -/- and LRRK2(G2019S), unlike +/+ (Fig. 1A). Also, LRRK2(D1994A) has dramatically reduced levels of LC3 and p62 (Fig. 1F). Is this just experimental variance or an indication of distinct effects on autophagy? I am not sure how exactly the autophagy experiments fit into to the MAM ER-mito flux story.

5) page 7, line 8: correct LRRK2(G2019S)

6) MTT conversion measures mitochondrial dehydrogenase activities and therefore is a general marker for cell proliferation/viability and not specific for apoptosis. Use appropriate terms on page 8.

Also do correct: (stress) decreased cell viability MORE STRONGLY and "...LRRK2 regulates viability responses to ER stress, whereby kinase-active LRRK2(G2019S) ENHANCES cellular vulnerability..."

7) page 21, line 15: do correct Fis1 (not Fos1)

As for the increase in Drp1, it might be worth checking the expression level of its ER binding

partner, INF2.

Referee #2:

This manuscript entitled "LRRK2 regulates ER-mitochondrial tethering through the PERK-mediated ubiquitination pathway" describes the role of LRRK2 in the regulation of ER-mitochondria tethering. The authors used CRISPR/Cas9 modified MEF cells to analyze the function of LRRK2. This new version of the manuscript addresses most issues this reviewer raised previously. However, there are some that remain.

Moreover, this reviewer is not specialized on mitochondria regulation by kinases and E3-ligases, and thus cannot judge if the data on this part are sound.

Major points:

1. The EM data are not really convincing. For instance Fig. 1A, images are of poor quality; the ultrastructure of the mitochondria does not seem to be similar in the four pictures presented: is there a mitochondria phenotype in some of the cell lines, or is there a problem of sample preservation?
2. Along the same line, on Fig. 2E, the ultrastructure of LRRK2 (D1994A) and LRRK2^{-/-} cells seems to be different, more particularly for mitochondria. Cristae seem to be poorly preserved. Is there an issue with magnification? The size of ER tubes and ribosomes does not appear to be the same in the four panels.
3. Quantification and statistical analysis of EM data was performed on the basis of the number of images. The number of independent samples on which the experiment was performed, and the number of cells analyzed was not specified. Should not statistical analyses be done on independent samples?

Minor points:

1. Some typos remain on the figures. For instance, Fig3E, 3F etc...
2. Check the colors Fig. 3C

Referee #3:

The authors made appreciable efforts to address the concerns raised by the previous version of their manuscript. They added additional data in support of a PERK-dependent mechanism by which the three ubiquitin-protein ligases under investigation regulate the degradation of mitofusin 2, thereby disrupting endoplasmic reticulum to mitochondria tethering. Their results suggest that LRRK2 regulates this mechanism by engaging into direct interaction with the ligases, which blocks their PERK-mediated phosphorylation and subsequent degradation of mitofusin 2. Intriguingly, this mechanism appears to be regulated by LRRK2 kinase activity, in the sense that LRRK2-dependent auto-phosphorylation disrupts binding to the ligases, allowing for their PERK-mediated activation. In this context, LRRK2 function seems to be exclusively associated with binding and negative regulation of the ligases, explaining why the deletion of the LRRK2 gene and the G2019S mutation have similar effects.

At present the main hypothesis of this work appears insufficiently supported by the data presented. There is no direct evidence that LRRK2, LRRK2G2019S and LRRK2D1994A indeed bind the ligases differently and that an autophosphorylation-dependent mechanism affects this binding. LRRK2-dependent *in vitro* phosphorylation and binding assays would be useful for this. As a complementary experiment, it should be shown that binding of LRRK2 to the ligases is indeed different in LRRK2, LRRK2G2019S and LRRK2D1994A cells and that the observed differences correlate with corresponding differences in the phosphorylation status of LRRK2 S1292 and in the activation status of the ligases (Figure 4). Does expression of increasing amounts of the LRRK2 ligase-binding domain (LRRK2-d1) in LRRK2^{-/-} or LRRK2G2019S cells counteract PERK-mediated activation of the ligases and ubiquitylation of Mfn2? Along this line, why do the d1 fragment and LRRK2 inhibitors have no effect on calcium peaks in LRRK2^{-/-} cells, in which supposedly a fraction of the LRRK2 protein should be active (Figures 3C and 5A)? This is surprising, taking into account the effect of LRRK2 inhibition on the phosphorylation status of the

ligases and the ubiquitylation status of mitofusin 2 in LRRK2^{-/-} cells (Figures 4B and S5B)?

Additional specific comments:

- Why do the inactive ligases increase calcium transfer in LRRK2^{-/-} and G2019S cells ? Does this mean that the ligases have dominant negative effects over the wild type proteins (Figures 3E and S4C)? Is this expected based on what we know of the biology of these proteins ?
- The analysis of autophagy fluxes remains unclear (Figures 1F and 6B). In particular, it is unclear whether and how the authors took into account the effect of bafilomycin in the quantitative analysis shown in Figure 1F. Does this analysis correspond to the bafilomycin (-) condition, or is it a ratio between Bafilomycin (+) and (-) conditions ?
- For more clarity, Figure 3 C should probably be integrated into Figure 2. The main text is inconsistent with the figure and should be adapted (« A construct lacking ... lost its inhibitory effect on Ca⁺⁺ transfer , ... » should be changed into « A LRRK2 construct lacking... lost the ability to rescue Ca⁺⁺ transfer in LRRK2^{-/-} cells, ... »
- Supplementary Figure 4A : insert reference on the upper part of the graphs, where appropriate (LRRK2^{+/+}, LRRK2^{-/-} etc.)
- Page 8. Cell survival under stress, line 5 : should be « ... to a higher extent »
- Figure S8 : Error bars are missing
- A generally confusing aspect is that some figure panels are presented before preceding figures/panels (for example, Figure 3D is presented before Figures 3E and F). This situation should be avoided in general.

2nd Revision - authors' response

30th Jul 2019

Referee #1:

The manuscript has greatly improved by the identification of PERK as the kinase activating LRRK2-autophosphorylation liberated ubiquitin ligases. Now the findings and conclusions make much more sense overall. Please consider a few details:

1) The authors determine global serine phosphorylations of the ubiquitin ligases. At least for parkin, a stimulatory phosphorylation is known to occur at serine-65. Is this the residue targeted by PERK?

Parkin is phosphorylated by PINK1 at Serine 65, the position of which is N-terminal ubiquitin like domain upstream from the RING domain (Shiba-Fukushima, Imai et al., 2012) (Kondapalli, Kazlauskaite et al., 2012). In the Figure EV4B, we performed immunoprecipitation/immunoblotting of Parkin or Parkin(S65A) co-expressed with PERK or kinase-defective PERK(K618R) in HEK293 cells. Parkin(S65A) was hardly phosphorylated by PERK. Thus, at least for Parkin, S65 is the phosphorylation site for PERK. We added this point in Page 18, line 6-11

REFERENCES

Kondapalli C, Kazlauskaite A, Zhang N, Woodroof HI, Campbell DG, Gourlay R, Burchell L, Walden H, Macartney TJ, Deak M, Knebel A, Alessi DR, Muqit MM (2012) PINK1 is activated by mitochondrial membrane potential depolarization and stimulates Parkin E3 ligase activity by phosphorylating Serine 65. *Open Biol* 2: 120080
 Shiba-Fukushima K, Imai Y, Yoshida S, Ishihama Y, Kanao T, Sato S, Hattori N (2012) PINK1-mediated phosphorylation of the Parkin ubiquitin-like domain primes mitochondrial translocation of Parkin and regulates mitophagy. *Sci Rep* 2: 1002

2) page 3, line 8: correct domain names Ras of complex proteins (Roc) and C-terminal of Roc (COR) [Bosgraaf & Haastert (2003) BBA 1643:5]

In the revised text, we collected mistakes in Page 3, line 8-9.

3) double-check text annotation for Fig. S1D

In the revised text, we changed annotation as followed (Supplementary Information Page 1, line 19-23):

Peak values of Ca^{2+} transients in LRRK2^{-/-} MEFs transfected with LRRK2(D1994A) were higher than that those in LRRK2^{-/-} MEFs transfected with control vector, whereas those in LRRK2^{-/-} MEFs transfected with LRRK2(G2019S) were not significantly different from that those in LRRK2^{-/-} MEFs transfected with control vector.

4) D1994A behaves similar or even stronger than +/+ in contrast to -/- and G2019S in most of the experiments. However, LRRK2(D1994A) cells show the same electron-dense material as -/- and LRRK2(G2019S), unlike +/+ (Fig. 1A). Also, LRRK2(D1994A) has dramatically reduced levels of LC3 and p62 (Fig. 1F). Is this just experimental variance or an indication of distinct effects on autophagy? I am not sure how exactly the autophagy experiments fit into to the MAM ER-mito flux story.

In this study, LRRK2(D1994A) augments the ER-mitochondrial interaction through the stabilization of MAN tether proteins such as mitofusin2. As shown in Figure 1B, 1C, 1D, LRRK2(D1994A) increased mitochondrial energetics. As citrate synthase activity, a marker of mitochondrial contents, was increased in LRRK2(D1994A), the suppressive effect of LRRK2(D1994A) on autophagic flux (Figure 1F) is not contradictory to the promotive effect of LRRK2(D1994A) on mitochondrial biogenesis. However, as reviewer pointed out, EM images of LRRK2(D1994A) MEFs showed the debris in the cytoplasm (Figure 1A). Question is how these debris accumulated in LRRK2(D1994A) MEFs.

During the autophagic flux, autophagosome is transported and fused with lysosome, where the materials in autophagosomes are degraded. Accumulation of undigested debris in LRRK2(D1994A) MEFs may be due to the defect in the endosome-lysosome fusion step. In fact, phosphoproteomics have showed that a subset of small G-proteins are substrates for LRRK2 (Steger, Tonelli et al., 2016). Especially, Rab7, a key mediator for endosome-lysosome fusion, is phosphorylated by *drosophila* LRRK2 homologue (Dodson, Zhang et al., 2012). Therefore, we speculate that phosphorylation-defective LRRK2(D1994A) should disturb intracellular transport of late endosome containing damaged materials to lysosome.

In the revised text, we added this points in Page 6, line 10-18

REFERENCES

Dodson MW, Zhang T, Jiang C, Chen S, Guo M (2012) Roles of the *Drosophila* LRRK2 homolog in Rab7-dependent lysosomal positioning. *Hum Mol Genet* 21: 1350-63
 Steger M, Tonelli F, Ito G, Davies P, Trost M, Vetter M, Wachter S, Lorentzen E, Duddy G, Wilson S, Baptista MA, Fiske BK, Fell MJ, Morrow JA, Reith AD, Alessi DR, Mann M (2016) Phosphoproteomics reveals that Parkinson's disease kinase LRRK2 regulates a subset of Rab GTPases. *Elife* 5

5) page 7, line 8: correct LRRK2(G2019S)

In the revised text, we collected mistakes.

6) MTT conversion measures mitochondrial dehydrogenase activities and therefore is a general marker for cell proliferation/viability and not specific for apoptosis. Use

appropriate terms on page 8.

Also do correct: (stress) decreased cell viability MORE STRONGLY and "...LRRK2 regulates viability responses to ER stress, whereby kinase-active LRRK2(G2019S) ENHANCES cellular vulnerability..."

In the revised text, we used viability instead of apoptosis as reviewer pointed out in Page 8, line 21 and Page 9, line 4 and corrected words in Page 9, line 1 and 5.

7) page 21, line 15: do correct Fis1 (not Fos1) As for the increase in Drp1, it might be worth checking the expression level of its ER binding partner, INF2.

In the revised Figure 3A, Appendix Figure S1A, we performed immunoblot using anti-formin2 (formin2; mouse homologue to human INF2) antibody. Formin2 levels were almost similar among MEFs of indicated genotypes. Thus, ER-positioned proteins such as Bap31, VAPB and formin2 revealed not to be substrates for mitochondrial ubiquitin ligases. We corrected mis-spelling of Fis1 and added formin2 expression in MAM in Page 12, line 11.

Referee #2:

This manuscript entitled "LRRK2 regulates ER-mitochondrial tethering through the PERK-mediated ubiquitination pathway" describes the role of LRRK2 in the regulation of ER-mitochondria tethering. The authors used CRISPR/Cas9 modified MEF cells to analyze the function of LRRK2.

This new version of the manuscript addresses most issues this reviewer raised previously. However, there are some that remain.

Moreover, this reviewer is not specialized on mitochondria regulation by kinases and E3-ligases, and thus cannot judge if the data on this part are sound.

Major points:

1. The EM data are not really convincing. For instance Fig. 1A, images are of poor quality; the ultrastructure of the mitochondria does not seem to be similar in the four pictures presented: is there a mitochondria phenotype in some of the cell lines, or is there a problem of sample preservation?

In the revised Figure 1A, we increased contrast of EM images using photoshop as shown below. Although revised EM images may be not enough for the evaluation of mitochondrial morphology, images obtained by fluorescent confocal microscopy using Mitotracker staining show the morphological differences among LRRK2 mutant expressing MEFs (Figure EV1E). Thus, there were morphological differences in mitochondria among mutant LRRK2 MEFs. In the revised text, we mentioned this point in Page 6, line 18-20.

2. Along the same line, on Fig. 2E, the ultrastructure of LRRK2 (D1994A) and LRRK2^{-/-} cells seems to be different, more particularly for mitochondria. Cristae seem to be poorly preserved.

Is there an issue with magnification? The size of ER tubes and ribosomes does not appear to be the same in the four panels.

In the revised Figure 2E, we increased contrast of EM images using photoshop as shown below. Although revised EM images may appear to be improved, the detailed evaluation of mitochondrial morphology including cristae remains unclear. Regarding magnification of EM images, we carefully adjusted the EM images in the revised Figure 2E.

3. Quantification and statistical analysis of EM data was performed on the basis of the number of images. The number of independent samples on which the experiment was performed, and the number of cells analyzed was not specified. Should not statistical analyses be done on independent samples?

Five images, each of which contained 2-4 mitochondria, were obtained from 8 MEFs of indicated genotypes. In total, we analyzed 40 images of each group of MEFs. In the revised text, we added this information in Methods (Page 35, line 13-14).

Minor points:

1. Some typos remain on the figures. For instance, Fig3E, 3F etc...
2. Check the colors Fig. 3C

In the revised Figures, we carefully checked spellings and grammatical errors, and color used in bar graphs.

Referee #3:

The authors made appreciable efforts to address the concerns raised by the previous version of their manuscript. They added additional data in support of a PERK-dependent mechanism by which the three ubiquitin-protein ligases under investigation regulate the degradation of mitofusin 2, thereby disrupting endoplasmic reticulum to mitochondria tethering. Their results suggest that LRRK2 regulates this mechanism by engaging into direct interaction with the ligases, which blocks their PERK-mediated phosphorylation and subsequent degradation of mitofusin 2. Intriguingly, this mechanism appears to be regulated by LRRK2 kinase activity, in the sense that LRRK2-dependent auto-phosphorylation disrupts binding to the ligases, allowing for their PERK-mediated activation. In this context, LRRK2 function seems to be exclusively associated with binding and negative regulation of the ligases, explaining why the deletion of the LRRK2 gene and the G2019S mutation have similar effects.

At present the main hypothesis of this work appears insufficiently supported by the data presented. There is no direct evidence that LRRK2, LRRK2G2019S and LRRK2D1994A indeed bind the ligases differently and that an autophosphorylation-dependent mechanism affects this binding. LRRK2-dependent in

vitro phosphorylation and binding assays would be useful for this. As a complementary experiment, it should be shown that binding of LRRK2 to the ligases is indeed different in LRRK2, LRRK2G2019S and LRRK2D1994A cells and that the observed differences correlate with corresponding differences in the phosphorylation status of LRRK2 S1292 and in the activation status of the ligases (Figure 4).

In the revised Figure 6A, to determine whether PERK-phosphorylation of ubiquitin ligases was regulated by mutant LRRK2 through their binding to ubiquitin ligases, we performed the immunoprecipitation/immunoblotting of transfected cell with PERK and ubiquitin ligase in combination with mutant LRRK2. Levels of phosphorylated ubiquitin ligases were suppressed by LRRK2(D1994A) more strongly than LRRK2 or LRRK2(G2019S), while the levels of bound ubiquitin ligases to LRRK2(D1994A) were more than those to LRRK2 or LRRK2(G2019S). Thus, the PERK-phosphorylation of ubiquitin ligases revealed to be opposite to the binding to mutant LRRK2. In the revised text, we added comments in page 19, line 6-15.

As shown in revised Figure 6C, 6D (original Figure 5F, 6A) and mentioned in Page 19, line 22- Page 20, line 6, PERK-phosphorylation was enhanced by un-bound auto-phosphomimic LRRK2(S1292D), but was suppressed by bound auto-phospho-defective LRRK2(S1292A). Altogether, we concluded that LRRK2 phosphorylation state was correlated with PERK-phosphorylation of ubiquitin ligases.

Does expression of increasing amounts of the LRRK2 ligase-binding domain (LRRK2-d1) in LRRK2^{-/-} or LRRK2G2019S cells counteract PERK-mediated activation of the ligases and ubiquitylation of Mfn2?

In the revised Figure 5E (data not shown in original Figures), we performed immunoprecipitation/immunoblotting of transfected LRRK2(G2019S) MEFs transfected with the increasing amounts of LRRK2-d1, a potential binder to ubiquitin ligases, under tunicamycin treatment. The increasing amounts of LRRK2-d1 progressively suppressed the PERK phosphorylation of ubiquitin ligases and rescued mitofusin2 levels. We added this point in Page 19, line 1-4.

Along this line, why do the d1 fragment and LRRK2 inhibitors have no effect on calcium peaks in LRRK2^{-/-} cells, in which supposedly a fraction of the LRRK2 protein should be active (Figures 3C and 5A)?

LRRK2^{-/-} MEFs were generated by CRISPR/Cas9 system and the loss of LRRK2 was confirmed by immunoblot (Figure EV1A). When LRRK2^{-/-} MEFs were transfected with LRRK2-d1, LRRK2-d1 bound ubiquitin ligases and decreased PERK phosphorylation of ubiquitin ligases thereby increasing ER-mitochondria interaction and peak Ca transients (Figure 3C). While, when LRRK2^{-/-} MEF were treated with LRRK2-IN, an inhibitor for LRRK2 kinase, LRRK2^{-/-} MEFs did not show any changes in peak Ca transient because of lack of LRRK2 (Figure 5A). Thus, LRRK2-d1 function as dominant negative inhibitor for ubiquitin ligases. Therefore, the data in Figure 3C is not contradictory to that in Figure 5A.

This is surprising, taking into account the effect of LRRK2 inhibition on the phosphorylation status of the ligases and the ubiquitylation status of mitofusin 2 in LRRK2^{-/-} cells (Figures 4B and 5B)?

In the Figure EV3A, we performed immunoprecipitation/immunoblot of LRRK2^{-/-} MEFs under tunicamycin. LRRK2^{-/-} MEFs showed more phosphorylation of ubiquitin ligases, more ubiquitinated mitofusin2 and less amounts of mitofusin2, which appeared to be similar to changes observed in LRRK2(G2019S) MEFs. In the revised text, we added this point in Page 16, line 7 and 9.

Additional specific comments:

- Why do the inactive ligases increase calcium transfer in LRRK2^{-/-} and G2019S cells? Does this mean that the ligases have dominant negative effects over the wild type proteins (Figures 3E and S4C)? Is this expected based on what we know of the biology of these proteins?

MARCH5, MULAN and Parkin belong to the E3 ubiquitin ligase family. It is assumed that the regulatory mechanism of their activation is similar. The regulatory mechanism of parkin has been extensively studied, and we exemplify parkin. During the ubiquitination process, Parkin, the activity of which is regulated by PINK1-phosphorylation, binds to the E2-co-enzyme via its RING domain and it physically receives the ubiquitin moiety on its active center Cys431 (Caulfield, Fiesel et al., 2015). When cells over-expressed with parkin(C431A) are incubated with mitochondrial toxin CCCP, they do not show the parkin-catalyzed degradation of mitochondrial proteins, even though cells contain endogenous parkin (Riley, Loughheed et al., 2013). It is thus suspected that over-expressed parkin(C431A) binds to endogenous E2-co-enzyme thereby excluding access of endogenous parkin to E2-co-enzyme. In the revised text, we added this point in Page 13, line 10-12, and Page 14, line 7-10.

REFERENCES

- Caulfield TR, Fiesel FC, Springer W (2015) Activation of the E3 ubiquitin ligase Parkin. *Biochem Soc Trans* 43: 269-74
- Riley BE, Loughheed JC, Callaway K, Velasquez M, Brecht E, Nguyen L, Shaler T, Walker D, Yang Y, Regnstrom K, Diep L, Zhang Z, Chiou S, Bova M, Artis DR, Yao N, Baker J, Yednock T, Johnston JA (2013) Structure and function of Parkin E3 ubiquitin ligase reveals aspects of RING and HECT ligases. *Nat Commun* 4: 1982

- The analysis of autophagy fluxes remains unclear (Figures 1F and 6B). In particular, it is unclear whether and how the authors took into account the effect of bafilomycin in the quantitative analysis shown in Figure 1F. Does this analysis correspond to the bafilomycin (-) condition, or is it a ratio between Bafilomycin (+) and (-) conditions?

Bafilomycin A1 prevents maturation of autophagic vacuoles by inhibiting fusion between autophagosomes and lysosomes. Increased LC3 levels are derived from two mechanisms; increased autophagosome formation and/or decreased degradation of autophagosomes. Using bafilomycin enable to distinguish two mechanisms. LC3 levels in LRRK2(D1994A) was smaller than those in other LRRKs in control as well as bafilomycin treatment, suggesting that autophagic formation was suppressed in LRRK2(D1994A). While, LC3 levels in LRRK2(G2019S) was larger than other LRRK2s in control and bafilomycin treatment, suggesting the autophagic formation was enhanced in LRRK2(G2019S). Ratio of LC3 and p62 was calculated in the absence of bafilomycin. In the revised text, we described this point in Page 8, line 8-13.

- For more clarity, Figure 3 C should probably be integrated into Figure 2. The main text is inconsistent with the figure and should be adapted (« A construct lacking ... lost its inhibitory effect on Ca⁺⁺ transfer, ... » should be changed into « A LRRK2 construct lacking... lost the ability to rescue Ca⁺⁺ transfer in LRRK2^{-/-} cells, ... »

Figure 2 showed the role of LRRK2 in the ER-mitochondrial interaction, and Figure 3 showed physical interaction between LRRK2 and ubiquitin ligases. Based on the data shown in Figure 3A, 3B, 3C, we screened the interacting molecules with LRRK2, which potentially decreased MAM components. Thus, we think it reasonable that Figure 3A, 3B, 3C are combined with 3D, 3E, 3F.

In the revised text (Page 13, line 3-4), we changed explanation as followed; “construct lacking ... lost its inhibitory effect on Ca⁺⁺ transfer, ...” to “A LRRK2 construct lacking... lost the ability to rescue Ca transfer observed in LRRK2^{-/-} MEFs”.

- Supplementary Figure 4A: insert reference on the upper part of the graphs, where appropriate (LRRK2^{+/+}, LRRK2^{-/-} etc.)

In the Appendix Figure S1A (original Figure S4A), we added references.

- Page 8. Cell survival under stress, line 5: should be « ... to a higher extent »

In the revised text, we changed “to a lesser extent” to “more strongly” (Page 9, line 1)

- Figure S8: Error bars are missing

In the Figure EV4C (original Figure S8), we added error bars.

- A generally confusing aspect is that some figure panels are presented before preceding figures/panels (for example, Figure 3D is presented before Figures 3E and F). This situation should be avoided in general.

During the revision process, we added new results to original ones. Please allow us to change figure numbers in revised Figure 5, 6, 7. In the response to reviewers, we added old figure number in parenthesis following revised figure number, if original figure number is changed in the revised text.

3rd Editorial Decision

1st Oct 2019

Thank you for submitting your revised manuscript for consideration by The EMBO Journal. Please accept my sincere apologies for the unusual delay in getting back to you. Your amended study was sent back to one of the referees for re-evaluation, and we have received his/her comments, which I enclose below.

As you will see the referee finds that the concerns have been sufficiently addressed and is now in favour of publication, pending minor revision. Please note that we have editorially considered your response to the other referees and concluded that they have been addressed satisfactorily.

Thus, we are pleased to inform you that your manuscript has been accepted in principle for publication in The EMBO Journal, pending some minor issues related to the remaining discussion points of referee #3 as well as formatting and data representation as listed below, which need to be adjusted at re-submission.

REFEREE REPORTS:

Referee #3:

The authors clarified most of the concerns raised by their previous version. The new data regarding the relationship between the autophosphorylation status of LRRK2, its binding to and the PERK2-mediated phosphorylation of the E3 ligases, are convincing and now support the main conclusions drawn.

Regarding my previous points 3 and 4 (Along this line, why do the d1 fragment and LRRK2 inhibitors have no effect on calcium peaks in LRRK2^{-/-} cells, in which supposedly a fraction of the LRRK2 protein should be active (Figures 3C and 5A)? This is surprising, taking into account the effect of LRRK2 inhibition on the phosphorylation status of the ligases and the ubiquitylation status of mitofusin 2 in LRRK2^{-/-} cells (Figures 4B and S5B)?) my concern regarded the results for LRRK2^{+/+} cells and not LRRK2^{-/-} cells, as indicated. I am sorry for the confusion. It would be valuable if the authors could simply comment, where appropriate in the manuscript, on why in their opinion the d1 fragment and LRRK2 inhibitors have no effect on calcium peaks in LRRK2^{+/+} cells, which express the normal LRRK2 protein.

Minor point:

- The authors should indicate in the legend to Figure 1F that ratios in the graphs were calculated in the absence of bafilomycin.

3rd Revision - authors' response

24th Oct 19

Response to comments from Referee #3:

The authors clarified most of the concerns raised by their previous version. The new data regarding the relationship between the autophosphorylation status of LRRK2, its binding to and the PERK2-mediated phosphorylation of the E3 ligases, are convincing and now support the main conclusions drawn.

Regarding my previous points 3 and 4 (Along this line, why do the d1 fragment and LRRK2 inhibitors have no effect on calcium peaks in LRRK2^{-/-} cells, in which supposedly a fraction of the LRRK2 protein should be active (Figures 3C and 5A)? This is surprising, taking into account the effect of LRRK2 inhibition on the phosphorylation status of the ligases and the ubiquitylation status of mitofusin 2 in LRRK2^{-/-} cells (Figures 4B and S5B)?) my concern regarded the results for LRRK2^{+/+} cells and not LRRK2^{-/-} cells, as indicated. I am sorry for the confusion. It would be valuable if the authors could simply comment, where appropriate in the manuscript, on why in their opinion the d1 fragment and LRRK2 inhibitors have no effect on calcium peaks in LRRK2^{+/+} cells, which express the normal LRRK2 protein.

In this study, we found that the binding of LRRK2 to E3 ubiquitin ligases plays the key role in regulation of their ligase activities for target protein such as mitofusin2 and its binding affinity is shut-down by the autophosphorylation. We focus on the effects of LRRK2 inhibitors, LRRK-D1 and LRRK2-IN-1 on ER-mitochondrial calcium transfer, the outcome of the sequential signaling processes.

The ER-mitochondrial Ca^{2+} transfer in LRRK2(D2019S)-expressing MEFs, the level of which was the lowest, was significantly increased by over-expression of LRRK2-D1 as well as LRRK2-IN-1 (Figure 5A, Appendix Figure S1B), where LRRK2(D2019S), fully auto-phosphorylated, lacked the binding to E3 ubiquitin ligases (Figure 6A, 6B). While, the ER-mitochondrial Ca^{2+} transfer in LRRK2(D1994A)-expressing MEFs, the level of which was the highest, was not increased any more by two maneuvers (Figure 5A, Appendix S1), where the majority of LRRK2(D1994A), not auto-phosphorylated, constitutively bound to E3 ubiquitin ligases (Figure 6A, 6B). In comparison with mutant LRRK2-expressing MEFs, the ER-mitochondrial Ca^{2+} transfer in LRRK2^{+/+} MEFs, the level of which was middle, was partially increased by two maneuvers (Figure 3C, 5A), where a fraction of LRRK2, not auto-phosphorylated, constitutively bound to E3 ubiquitin ligases (Figure 6A (please see bottom autoradiograph), 6B (please see bottom panels)). Thus, the regulatory model of E3 ubiquitin ligases by LRRK2 could explain changes in the ER-mitochondrial Ca^{2+} transfer in mutant LRRK2-expressing MEFs.

In the revised manuscript, we added these sentences (page 26 lines 10 to page 27, line6).

Minor point:

- The authors should indicate in the legend to Figure 1F that ratios in the graphs were calculated in the absence of bafilomycin.

In the revised manuscript, we added “in the absence of bafilomycin” in Figure Legends of Figure 1F.

4th Editorial Decision

5th Nov 2019

Thank you for submitting the revised version of your manuscript. I have now evaluated your amended manuscript and concluded that the remaining minor concerns have been sufficiently addressed.

Thus, I am pleased to inform you that your manuscript has been accepted for publication in the EMBO Journal.

Corresponding Author Name: Toshihiko Toyofuku

Journal Submitted to: The EMBO Journal

Manuscript Number: EMBOJ-2018-100875